# Impact of El Niño Southern Oscillation on the interannual variability of methane and tropospheric ozone

Matthew J. Rowlinson[1], Alexandru Rap[1], Stephen R. Arnold[1], Richard J. Pope[1,2], Martyn P. Chipperfield[1,2], Joe McNorton[3], Piers Forster[4], Hamish Gordon[1], Kirsty J. Pringle[1], Wuhu Feng[1,5], Brian J. Kerridge[6,7], Barry L. Latter[6,7], Richard Siddans[6,7]

[1]Institute for Climate and Atmospheric Science, School of Earth and Environment, University of Leeds, Leeds, LS2 9JT, UK
[2]National Centre for Earth Observation, University of Leeds, Leeds, LS2 9JT, UK
[3]European Centre for Medium-Range Weather Forecasts, Reading, RG2 9AX, UK
[4]Priestley International Centre for Climate, University of Leeds, LS2 9JT, Leeds, UK
[5]National Centre for Atmospheric Science, University of Leeds, LS2 9JT, Leeds, UK
[6]Remote Sensing Group, STFC Rutherford Appleton Laboratory, Harwell, Oxfordshire, UK
[7]National Centre for Earth Observation, Harwell, Oxfordshire, UK

*Correspondence to*: Matthew J. Rowlinson (ee11mr@leeds.ac.uk)

**Abstract.** The interannual variability of greenhouse gases methane ($CH_4$) and tropospheric ozone ($O_3$) is largely driven by natural variations in global emissions and meteorology. The El Niño Southern Oscillation (ENSO) is known to influence fire occurrence, wetland emission and atmospheric circulation, affecting sources and sinks of $CH_4$ and tropospheric $O_3$, but there are still important uncertainties associated with the exact mechanism and magnitude of this effect. Here we use a modelling approach to investigate how fires and meteorology control the interannual variability of global carbon monoxide (CO), $CH_4$ and $O_3$ concentrations, particularly during large El Niño events. Using a three-dimensional chemical transport model (TOMCAT) coupled to a sophisticated aerosol microphysics scheme (GLOMAP) we simulate changes to CO, hydroxyl radical (OH) and $O_3$ for the period 1997-2014. We then use an offline radiative transfer model to quantify the climate impact of changes to atmospheric composition as a result of specific drivers.

During the El Niño event of 1997-1998, there were increased emissions from biomass burning globally, causing global CO concentrations to increase by more than 40%. This resulted in decreased global mass-weighted tropospheric OH concentrations of up to 9% and a consequent 4% increase in the $CH_4$ atmospheric lifetime. The change in $CH_4$ lifetime led to a 7.5ppb $yr^{-1}$ increase in the global mean $CH_4$ growth rate in 1998. Therefore, biomass burning emission of CO could account for 72% of the total effect of fire emissions on $CH_4$ growth rate in 1998.

Our simulations indicate that variations in fire emissions and meteorology associated with El Niño have opposing impacts on tropospheric $O_3$ burden. El Niño-related changes in atmospheric transport and humidity decrease global tropospheric $O_3$ concentrations leading to a -0.03 $Wm^{-2}$ change in the $O_3$ radiative effect (RE). However, enhanced fire emission of precursors such as nitrogen oxides ($NO_x$) and CO increase $O_3$ and lead to an $O_3$ RE of 0.03 $Wm^{-2}$. While globally the two mechanisms nearly cancel out, causing only a small change in global mean $O_3$ RE, the regional changes are large – up to -0.33 $Wm^{-2}$ with potentially important consequences for atmospheric heating and dynamics.

## 1 Introduction

In terms of radiative forcing, methane ($CH_4$) is the second most important anthropogenically emitted greenhouse gas after $CO_2$ (Myhre et al., 2013). Concentrations of $CH_4$ have risen from approximately 722 ppb in 1750 to over 1850 ppb in 2018, an increase of more than 150% (Dlugokencky, 2019). During this time period $CH_4$ has contributed an estimated radiative forcing (RF) of $0.48 \pm 0.05$ Wm$^{-2}$, around 20% of the total direct anthropogenic RF from greenhouse gases (Myhre et al., 2013). Furthermore, $CH_4$ is a precursor of tropospheric ozone ($O_3$), which is also a greenhouse gas responsible for a RF of $0.4 \pm 0.2$ Wm$^{-2}$ since the pre-industrial (Myhre et al., 2013), as well as a harmful pollutant that damages human health (Anenberg et al., 2010) and ecosystems (Sitch et al., 2007). While anthropogenic emissions have driven the long-term increase in $CH_4$ concentrations, $CH_4$ is also emitted from a range of natural sources leading to strong interannual variability (IAV) (Bousquet et al., 2006; Dlugokencky et al., 2011; Nisbet et al., 2016). Understanding the mechanisms driving IAV is important for accurate predictions of future $CH_4$ concentrations, especially in the context of anthropogenic emission reductions.

Previous studies indicate that although anthropogenic sources may contribute to seasonal variations in atmospheric $CH_4$, natural sources are the primary drivers of IAV (Bousquet et al., 2006; Meng et al., 2015). Emissions from natural wetlands have been shown to be the dominant process, with emissions from fires and changes to the atmospheric sink also playing important roles (Bousquet et al., 2006; Chen and Prinn, 2006; Dlugokencky et al., 2011; Kirschke et al., 2013; McNorton et al., 2016a; Corbett et al., 2017; McNorton et al., 2018). These natural sources are climate sensitive, so interannual changes to temperature and precipitation affect the amount of $CH_4$ emitted into the atmosphere as well as the spatial distribution (Zhu et al., 2017). A number of studies have found that biomass burning emissions are largely responsible for the IAV of carbon monoxide (CO) and also affect $O_3$ concentrations (Granier et al., 2000; Monks et al., 2012; Voulgarakis et al., 2015); however Szopa et al. (2007) suggested that meteorology is a more important driver of IAV for CO, explaining 50-90% of IAV.

A major driver of climatic IAV is the El Niño Southern Oscillation (ENSO) – a mode of climate variability originating in the Pacific Ocean with alternating warm (El Niño) and cold (La Niña) modes (McPhaden et al., 2006). Positive phase El Niño events lead to warmer and drier conditions in much of the tropics, disrupting global circulation patterns and leading to widespread changes in fire occurrence, wetland emissions and atmospheric transport (Feely et al., 1987; Jones et al., 2001; McPhaden et al., 2006). These influences occur most strongly in the tropics but have global consequences (Jones et al., 2001).

Global $CH_4$ concentrations have been observed to increase significantly during El Niño events, with an especially strong signal during the 1997-1998 event when the $CH_4$ growth rate was 12 ppb yr$^{-1}$, almost triple the 1750-2018 mean annual growth rate (Rigby et al., 2008; Hodson et al., 2011). Due to the wide-ranging effects of El Niño and varied sources of $CH_4$, there are multiple factors which could trigger the increase in $CH_4$ growth rate. Chen and Prinn (2006) attributed the increase to anomalies in global wetland emissions; however Zhu et al. (2017) estimated that although 49% of the interannual variation in wetland emissions can be explained by ENSO, wetland emissions were significantly lower during El Niño including the 1997-1998 event. Conversely, Schaefer et al. (2018) estimated that ENSO is responsible for up to 35% of global $CH_4$ variability, but the effect wetland and biomass burning emission changes are dwarfed by processing affecting the OH sink. Bousquet et al. (2006) suggested that the increased $CH_4$ growth rate during the 1997-1998 El Niño was primarily caused by abnormally large peat fires in Indonesia emitting huge amounts of $CH_4$ while wetlands emissions remained stable (van der Werf et al., 2004; Butler et al., 2005; Bousquet et al., 2006).

In addition to direct emissions of $CH_4$ from fires, it has been proposed that anomalously large CO emissions during enhanced El Niño fire events could explain the changes to $CH_4$ growth rate (Butler et al., 2005; Bousquet et al., 2006). CO is emitted from biomass burning in much larger quantities than $CH_4$ (~20× larger) and its reaction with the hydroxyl radical (OH) is its primary atmospheric sink (Voulgarakis and Field, 2015). Abnormal increases in CO concentrations may suppress the availability of OH, thereby extending $CH_4$ lifetime and increasing its growth rate during and following large fire events (Butler et al., 2005; Manning et al., 2005). The reaction of $CH_4$ with OH is the largest term in the global $CH_4$ budget, accounting for

~90% of its sink (McNorton et al., 2016a), therefore even minor changes to OH caused by the presence of other compounds or changes to atmospheric transport and photolysis rates could have a large impact on $CH_4$ growth rate (Dlugokencky et al., 2011). Butler et al. (2005) found that CO emissions suppressed OH concentrations by 2.2% in 1997-1998, which accounted for 75% of the observed change in $CH_4$ concentration. Bousquet et al. (2006) also reported a weakened OH sink during this El Niño event.

Here we use a modelling approach to investigate how El Niño events affect global $CH_4$, CO and tropospheric $O_3$ concentrations through changes to fire occurrence and atmospheric transport. Using long-term simulations spanning multiple El Niño and La Niña events, we quantify the relative influence of changes to fire emissions and dynamical transport. We also differentiate between the effect of direct $CH_4$ emissions from fires and the indirect effect via CO emissions and atmospheric chemistry changes.

## 2 Models and Simulations

### 2.1 Model description

For this study we use the TOMCAT chemical transport model (Chipperfield, 2006) coupled to the GLOMAP global aerosol microphysics scheme (Mann et al., 2010). The version of TOMCAT-GLOMAP used here is a further development of that described by Monks et al. (2017). Cloud fields are now provided from the European Centre for Medium-Range Weather Forecasts (ECMWF) reanalyses (Dee et al., 2011), replacing the climatological clouds fields used previously from the International Satellite Cloud Climatology Project (ISCCP) (Rossow and Schiffer, 1999), leading to improved representation of photolysis. Other developments include updated emission inventories, the inclusion of CERN CLOUD-based new particle formation and the introduction of Martensson sea spray emissions (Gordon et al., 2017; Monks et al., 2017). The model is run at $2.8° \times 2.8°$ horizontal resolution with 31 vertical levels from the surface to 10 hPa, driven by 6-hourly ECMWF ERA-Interim reanalyses. The planetary boundary layer (PBL) scheme is based on Holtslag and Boville (1993) and sea surface temperatures are from ECMWF reanalyses. ECMWF ERA-Interim reanalyses have been shown to have good skill in capturing Madden-Julian Oscillation (MJO) events which in turn impact the onset of ENSO events (Dee et al., 2011), giving confidence that the model competently simulates El Niño meteorological conditions.

The tropospheric chemistry scheme used is as described in Monks et al. (2017) with anthropogenic emissions from the Monitoring Atmospheric Composition and Climate (MACCity) emissions inventories (Lamarque et al., 2010). Annually varying emission inventories are included for all fire-emitted gas-species and aerosol emissions such as black carbon (BC). The Global Fire Emissions Database (GFED) used by TOMCAT-GLOMAP has been updated to version 4 with CO, nitrogen oxides ($NO_x$) and volatile organic compound (VOC) emissions from fires (Randerson et al., 2017; Reddington et al., 2018). Monthly varying biogenic VOC emissions are from the MEGAN-MACC emissions inventory for reference year 2000, calculated from the Model of Emissions of Gases and Aerosols from Nature (MEGANv2) (Sindelarova et al., 2014). The $CH_4$ inventory was produced by McNorton et al. (2016b), with wetland emissions derived from the Joint UK Land Environment Simulator (JULES) and biomass burning emissions from GFEDv4 (Randerson et al., 2017). These are then combined with anthropogenic emissions from EDGARv3.2, paddy field emissions from Yan et al. (2009) and termite, wild animal, mud volcano, hydrate and ocean emissions from Matthews and Fung (1987) (McNorton et al., 2016b). The global mean surface $CH_4$ mixing ratio is scaled in TOMCAT-GLOMAP to a best-estimate based on observed global surface mean concentration (McNorton et al., 2016a; Dlugokencky, 2019).

### 2.2 Radiative transfer model

Radiative effects of $O_3$ changes are calculated using the $O_3$ radiative kernel approach derived by Rap et al. (2015) using an offline version of the Edwards and Slingo (1996) radiative transfer model. This considers six bands in the shortwave (SW),

nine bands in the longwave (LW) and uses a delta-Eddington two-stream scattering solver at all wavelengths (Rap et al., 2015). This version has been used extensively in conjunction with TOMCAT-GLOMAP for calculating radiative forcing from simulated distributions of several short-lived climate pollutants (SLCPs) including BC, $O_3$ and $CH_4$ (Spracklen et al., 2011; Riese et al., 2012; Rap et al., 2013; Richards et al., 2013; Rap et al., 2015).

## 2.3 Simulations

All simulations are performed for 1997-2014 with a four-year spin-up through 1993-1996. The control run (CTRL) allows all emissions and meteorology to vary throughout the modelled period. GFED biomass burning emission inventories began in 1997, therefore the 1993-1996 spin-up simulation uses repeating 1999 emissions instead, as the closest year of 'average' emissions, having excluded 1997 and 1998 due to the exceptionally high emissions in those years (Schultz et al., 2008).

To test the impact of El Niño events on atmospheric chemistry, we also performed 3 perturbed simulations (Table 1). Where model simulations used "Fixed" parameters in Table 1, the year 2013 emissions or meteorology are specified as invariant throughout the simulation. This year is chosen as the ENSO-neutral case, due to it being the least active ENSO year during 1997-2014, with a maximum bimonthly multivariate ENSO index (MEI) magnitude of -0.4 and the only year without a single MEI value that could be considered an active El Niño or La Niña (Wolter and Timlin, 1993; Wolter and Timlin, 1998). Throughout this study, an El Niño event was taken to be ongoing if the MEI was greater than +1.0. We perform a factorial analysis based on perturbed simulations in which we fix global biomass burning emissions (FIREFIX) or global meteorology (METFIX) to the 'ENSO-neutral' case. An additional perturbed simulation was performed in order to examine the secondary impact of CO on $CH_4$ via oxidation changes, where only CO emissions from biomass burning were fixed (COFIX).

**Table 1. Details of TOMCAT model simulations. All simulations are run for 1997-2014.**

| Simulation name | Meteorology | CO biomass burning emissions | All other biomass burning emissions |
|---|---|---|---|
| **CTRL** | Varying | Varying | Varying |
| **METFIX** | Fixed | Varying | Varying |
| **FIREFIX** | Varying | Fixed | Fixed |
| **COFIX** | Varying | Fixed | Varying |

## 3 Model Evaluation

We have conducted a comprehensive evaluation of the coupled TOMCAT-GLOMAP model using aircraft observations, and data from ozone sondes and satellites. In general, the model is able to capture absolute concentrations, global distribution and seasonal variations of major species including $O_3$, CO and $CH_4$. MOPITT satellite retrievals have been used to evaluate CO at 800 hPa and 500 hPa (Emmons et al., 2004) and are shown in Fig. S1 and S2, respectively, along with a description of the satellite product and the averaging kernels applied to the model output. TOMCAT performs similarly here as in Monks et al. (2017), underestimating CO concentrations in the Northern Hemisphere (NH) while overestimating peak concentrations in biomass burning regions, with a maximum difference of ~75ppb (Fig. S1 and S2). However, TOMCAT is able to reproduce seasonal variations in CO and locates peak CO accurately over East Asia and Central Africa.

Simulated $O_3$ concentrations from TOMCAT were also compared with satellite observations of lower tropospheric (0-6km) $O_3$ from the Ozone Monitoring Instrument (OMI). These data were provided by the Rutherford Appleton Laboratory (RAL; data version fv0214) using an optimal estimation retrieval scheme which resolves $O_3$ in the 0-6 km layer by exploiting information in the Hartley and Huggins UV bands. The scheme derives from that discussed by Miles et al. (2015) for another UV sounder GOME-2. TOMCAT representation of $O_3$ concentrations between 0-6 km in NH winter are slightly improved on

the Monks et al. (2017) version, particularly in tropical and Southern Hemisphere (SH) concentrations (Fig. S3). However, there remains a general low bias in global $O_3$ of up to 10 Dobson Units (DU) in winter in regions such as the southern Atlantic Ocean.

TOMCAT $O_3$ has also been evaluated using sonde observations (Fig. 1 and Fig. S4) (Tilmes et al., 2012), with the model generally representing the vertical profiles, seasonal variation and absolute concentrations of $O_3$ very well, with a normalised mean bias (NMB) of 1.1% across all sites at 700-1000 hPa and 2.1% at 300-700 hPa. The model capably simulates the seasonality of tropospheric $O_3$ (Fig. 1), with a maximum seasonal bias of 6.3% at 300-700 hPa in March-May. There is no apparent regional or latitudinal bias, although simulated concentrations are over-estimated in India (Fig. S4). In addition, the

TOMCAT-simulated global tropospheric burden of $O_3$ in 2000 is 342 Tg which falls within the range of published values (Table 2).

We have also assessed the capability of TOMCAT-GLOMAP to simulate observed responses to El Niño events. Ziemke et al. (2010) derived an $O_3$ ENSO index using satellite observations, finding that for a +1K change in the Nino 3.4 index, there was a 2.4 DU increase in the OEI. In TOMCAT-GLOMAP, we calculate a 2.8 DU increase per +1K in the Nino 3.4, indicating a

slightly larger but comparable response to El Niño events. The regional response of tropospheric $O_3$ to El Niño was evaluated against an analysis using various observations and a chemistry-climate model in Zhang et al. (2015). That study observed increased total $O_3$ column in the North Pacific, southern USA, north-eastern Africa and East Asia, with decreases over central Europe and the North Atlantic. All of these observed responses were present in TOMCAT-GLOMAP simulations, except with a slight increase in TOC in central Europe and a simulated decrease in Western Europe and East Atlantic (Fig. S5).


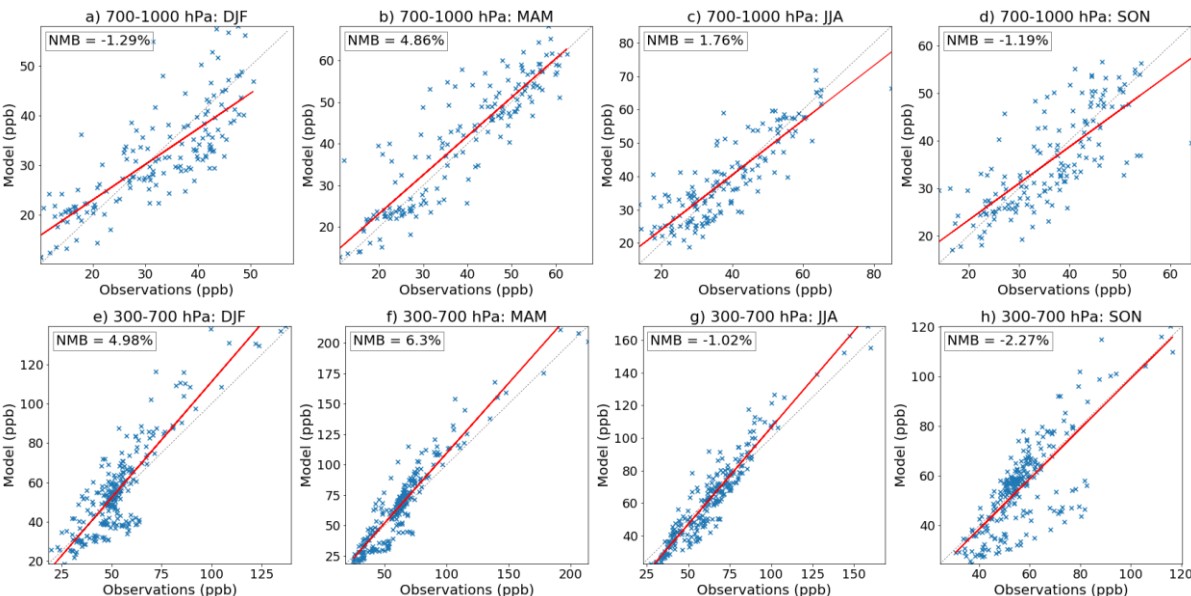

**Figure 1: Comparison of seasonal mean simulated $O_3$ concentrations (ppb) against mean ozonesonde observations from Tilmes et al. (2012), for the period 1995-2011. Panels a-d show mean concentrations at 700-1000 hPa across all sites, while panels e-h show mean concentrations at 300-700 hPa. Values in each panel are seasonal means, from left to right, December-February (DJF), March-May (MAM), June-August (JJA) and September-November (SON). The red line represents the linear regression. Normalised mean bias (NMB) values between model and observations are also shown.**

### 3.1 Aircraft observations

We compare annual mean simulated gas-phase species for 1999 against a climatological dataset of aircraft observations from 16 campaigns conducted from 1992 to 2001, with a broad spatial and temporal range (Emmons et al., 2010). While the comparison of observational data from intermittent aircraft campaigns does not offer a perfect comparison with the model

simulated long-term mean concentrations, it allows evaluation of broad characteristics of a number of species over vertical

profiles in many global regions. Figure 2 shows the comparison of simulated annual mean global concentrations of CO, $CH_4$ and PAN, with aircraft observations at 0- 2 km, 2-6 km and 6-10 km. We have also calculated the normalised mean bias between the model and observations (Fig. S6). Full details of the aircraft measurement campaigns used can be found in the Supplementary Table S1.

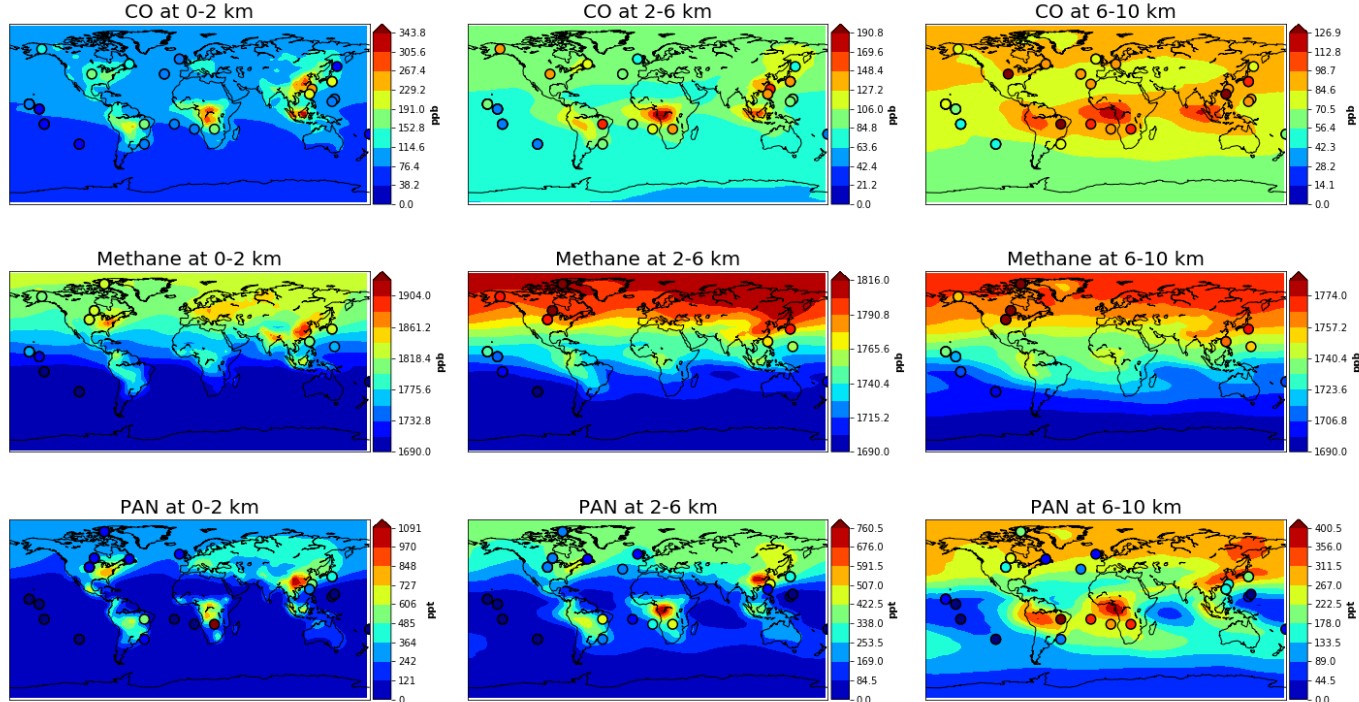

**Figure 2: Global mean volume mixing ratios of CO (ppb), $CH_4$ (ppb) and PAN (ppt) from TOMCAT for the period 1993-2001 at 0-2 km (left panels), 2-6 km (middle panels) and 6-10 km (right panels). The filled circles show mean values from aircraft observation campaigns which took place between 1992 and 2001 (Table S1) (Emmons et al., 2010).**


The model captures broad characteristics of spatial distribution for all species, simulating higher concentrations in polluted urban or biomass burning regions, with lower concentrations over ocean and in the SH. CO concentrations decrease with altitude but the largest values still occur around urban areas and burning regions, which can be seen in both model and aircraft concentrations. Consistent with the comparison with MOPITT satellite retrievals (Fig. S1 and S2), the model underestimates

CO concentrations particularly near the surface, with a NMB of -11.1%, -9.93% and -0.25% at 0-2 km, 2-6 km and 6-10 km, respectively. Absolute concentrations of $CH_4$ in TOMCAT simulations match aircraft data very well, although given the global mean surface concentration scaling we expect the magnitude of $CH_4$ to be well simulated. The latitudinal and vertical distributions are also well captured, giving confidence in the model transport and OH simulation. Aircraft observations show $CH_4$ also decreases with altitude and the hemispherical disparity becomes more pronounced, with higher concentrations in the

NH. For PAN concentrations, the simulated spatial distribution is broadly well captured, as is the increased concentration with altitude. There is a general low bias in absolute concentrations near the surface (NMB=-12.3%), with better comparison at 2-6 km (NMB=1.68%) and over-estimation at 6-10 km (NMB=18.17%).

**3.2 OH Evaluation**

Due to its very short lifetime, it is challenging to evaluate model-simulated OH over representative spatial and temporal scales.

Here we follow the evaluation methodology recommended by Lawrence et al. (2001) of dividing tropospheric OH into 12 sub-domains, from the surface to a climatologically derived tropopause. This method was also used to evaluate a previous version of TOMCAT(vn1.76) by Monks et al. (2017), allowing direct comparison. The evaluation is performed for the year 2000.

Figure 3 shows our simulated OH compared to Monks et al. (2017), the ACCMIP model mean (Naik et al., 2013) and the Spivakovsky et al. (2000) OH dataset estimated from methyl chloroform observations.

The models and observationally constrained distribution broadly agree in terms of the latitudinal spread of OH concentrations with a minimum in the SH and a maximum at the tropics; however there is disagreement over the exact altitude of the maximum OH concentrations. In both versions of TOMCAT the highest concentration is between the surface and 750hPa, while ACCMIP and Spivakovsky et al. (2000) find peak OH in the upper and mid-level troposphere, respectively. The updated cloud fields used in the current TOMCAT-GLOMAP version have slightly increased OH concentrations in the mid-level and upper

domains compared to Monks et al. (2017) but concentrations remain significantly higher in the NH and surface domains than in other studies. In addition, our simulated NH:SH ratio of 1.48 in the current TOMCAT version remains substantially higher than in the ACCMIP models (1.28 ± 0.1), indicating that TOMCAT photolysis rates and OH production in the NH are larger. The total global tropospheric average OH in this version of TOMCAT is $1.04 \times 10^6$ molecules cm$^{-3}$, a decrease from Monks et al. (2017) and within the range of other published values (Table 2). This is primarily due an updated treatment of clouds, in

which climatological cloud fields have been replaced with cloud fraction from ECMWF reanalyses data, affecting photolysis rates. The tropospheric O$_3$ burden of 342 Tg has increased relative to Monks et al. (2017) (331 Tg) and is within the range found in Wild (2007) (335 ± 10 Tg) and ACCMIP models (337 ± 23 Tg) (Young et al., 2013). Due to the simplified treatment of CH$_4$, the scaling applied and its relatively long atmospheric lifetime, the total atmospheric lifetime cannot be determined from TOMCAT simulations. Instead a chemical lifetime due to reaction with OH is calculated from CH$_4$ and OH burdens,

disregarding stratospheric and soil sinks (Fuglestvedt et al., 1999; Berntsen et al., 2005; Voulgarakis et al., 2013). The lifetime diagnosed from TOMCAT is 8.0 years, compared to the multi-model mean and range of 9.3 ± 0.9 years from Voulgarakis et al. (2013). The shorter lifetime in TOMCAT is due to the overestimation of OH at the surface, particularly in the NH where CH$_4$ concentrations are highest due to anthropogenic emissions.

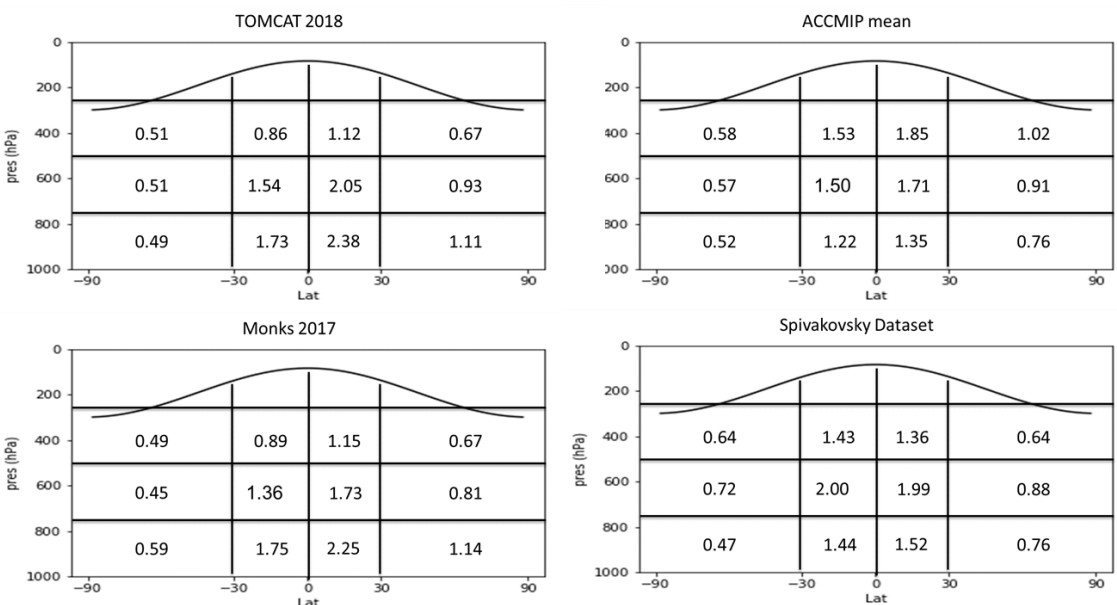

**Figure 3: Annual zonal mean hydroxyl radical (OH) concentrations ($\times 10^6$ molecules cm$^{-3}$) divided into 12 sub-domains as recommended by Lawrence et al. (2001). The simulated OH from this study is compared to a dataset estimated from methyl chloroform observations (Spivakovsky et al., 2000) and the Atmospheric Chemistry and Climate Model Intercomparison Project (ACCMIP) multi-model mean (Naik et al., 2013). Results from a previous version of TOMCAT from Monks et al. (2017) are also shown. A climatological tropopause, indicated by the smooth black line**
**near the top of each panel, has been used to remove stratospheric OH.**


**Table 2: Present day (2000) TOMCAT model diagnostics compared to previous model version from Monks et al., (2017) and other published values.**

| Diagnostic | TOMCAT (this study) | Monks et al. (2017) | Other estimates | Reference |
|---|---|---|---|---|
| $O_3$ burden (Tg) | 342 | 331 | $337 \pm 23$ | Young et al. (2013) |
| Tropospheric OH concentration ($\times 10^6$ molecules cm$^{-3}$) | 1.04 | 1.08 | 0.94-1.06 | Prinn et al. (2001); Krol and Lelieveld (2003); Bousquet et al. (2005); Wang et al. (2008). |
| $CH_4$ lifetime (years) | 8.0 | 7.9 | $9.3 \pm 0.9$ | Voulgarakis et al. (2013) |

## 4 Results and discussion

### 4.1 Impact of meteorology and fire emissions on trace gas interannual variability

First we examine the mechanisms controlling interannual variability of simulated tropospheric CO, $O_3$ and mean OH. We use the difference between the control (CTRL) and the perturbed simulations with fixed fires (FIREFIX) and fixed meteorology (METFIX) to determine the driving cause of IAV. Of particular interest is the effect of the 1997-1998 El Niño event (henceforth referred to as 1997 El Niño) and how the prevailing mechanisms controlling IAV change during such events. To define El Niño events, we use the bimonthly multivariate ENSO index, which is calculated from 6 observed variables and standardised to accurately monitor ENSO occurrence (Wolter and Timlin, 1998; Wolter and Timlin, 2011).

Previous studies examining the dominant factor controlling global CO IAV have found contrasting results. Szopa et al. (2007) suggested that meteorology was the main driver, accounting for 50-90% of IAV in the tropics. Conversely, a study by Monks et al. (2012) considered CO IAV in the Arctic, finding that biomass burning was the dominant driver with a strong correlation to El Niño. Voulgarakis et al. (2015) also suggested that biomass burning was the more important driver of IAV with only a small effect from meteorology. Some of these differences in results can be explained by Szopa et al. (2007) considering only surface CO rather than the whole troposphere as in Voulgarakis et al. (2015). Here we also consider whole tropospheric CO and our results are in line with those from Voulgarakis et al. (2015). We find the dominant source of IAV across the entire period is emissions from biomass burning - indicated by the large difference between simulations CTRL and FIREFIX (Fig. 4a), with a small effect from meteorological changes (CTRL – METFIX). This effect is largest during the 1997 El Niño where an increase in fire events increased CO concentrations by more than 40%. Smaller increases of 5.8% and 7.6% occur during less extreme El Niño events of 2002/2003 and 2006, respectively, with only a 1.8% increase during the 2009/2010 El Niño, indicating that El Niño only significantly impacts CO concentrations when there is an associated increase in global fire events. Expanding on the work of Voulgarakis et al. (2015), we analysed IAV using the coefficient of variation (CV) calculated as the multi-year standard deviation normalised by the mean (Fig. 5). The global annual mean CO IAV over the whole period is 11.0% for the whole troposphere and 14.3% for surface concentrations. This is in very good agreement with Voulgarakis et al. (2015) who calculated 10% IAV, in fact the comparison is even better when we consider the same time period (2005-2009), with our corresponding IAV estimate at 9.7%. The slightly lower estimate here may be a result of the fixed-year BVOC emissions, removing the effect of IAV of biogenic emissions on CO IAV. BVOC oxidation is estimated to contribute 15% of the total source of CO (Duncan et al., 2007), however the IAV of BVOC emissions has been found to be relatively small, ~2-4% (Naik et al., 2004; Lathière et al., 2005). Despite good global comparison with (Voulgarakis et al., 2015), there are regional differences; CO IAV from TOMCAT is much larger in high-latitude boreal regions. This is likely due to the difference in period studied meaning this study includes additional extreme events including unusually large Russia boreal wildfires in 2010

and 2012 (Gorchakov et al., 2014; Kozlov et al., 2014). Infrequent and extreme events such as these significantly increase IAV.

CO IAV is significantly greater in September-October, with peaks in known fire regions such as tropical South America, Africa, Southeast Asia and boreal forests. This indicates a strong contribution of fire emissions to IAV especially from Indonesia (Fig. 4a), as also suggested by previous studies (Monks et al., 2012; Huang et al., 2014; Voulgarakis et al., 2015). In the FIREFIX simulation IAV is ~55% of the CTRL value showing a large reduction in variability when interannual variability in fire emissions is removed. The IAV in March-April is significantly smaller than September-October as this period is outside the primary fire season for South America and Eurasia, although hotspots remain in Southeast Asia and Africa where fires commonly occur in March-April (van der Werf et al., 2017). Meteorology and atmospheric transport changes are most important in Africa in September-October and Indonesia in March-April (Fig. 5c, d). Fire emissions occur in these regions but the meteorological effects are important sources of IAV. This is in good agreement with Voulgarakis et al. (2015) who found that with fixed biomass burning emissions, there remained high IAV over Africa during Dec-Jan, and Huang et al. (2014) who found CO over Central Africa correlated more closely with ice water content than CO emissions due to increased convective transport. However, the overall effect of meteorology on global IAV found here is much smaller than the 50-90% suggested

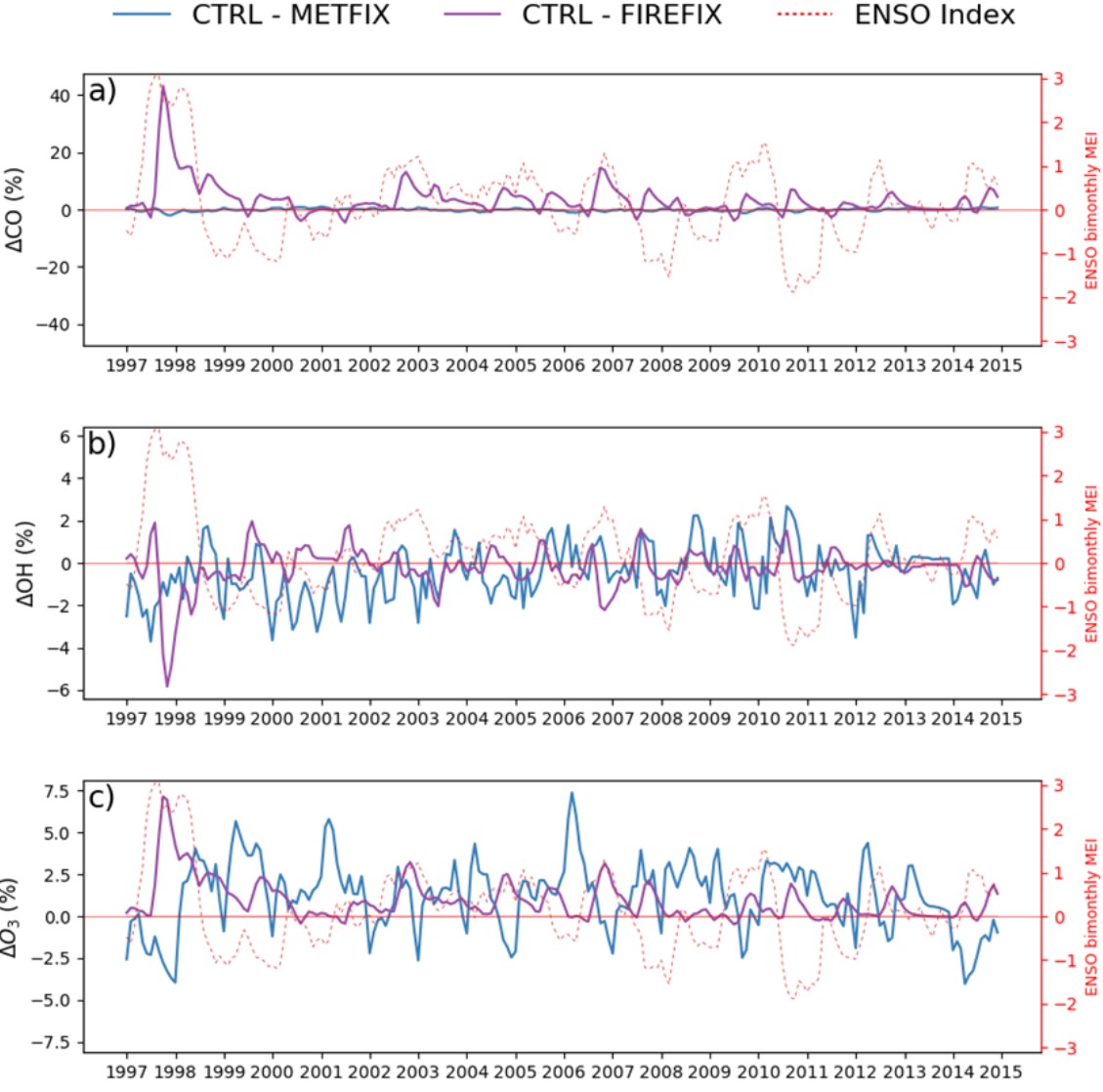

**Figure 4: Time series of simulated differences (%) between the control and the fixed meteorology (CTRL - METFIX, blue line) and fixed fire emissions (CTRL – FIREFIX, purple line) simulations for the global tropospheric burden of (a) CO, (b) OH and (c) $O_3$. The ENSO bimonthly mean multivariate index is plotted in the dashed red line on the right-hand y axis in each panel.**

by Szopa et al. (2007): when we consider only surface CO over the same period, fixing meteorology decreases the mean CO IAV by just 5%.

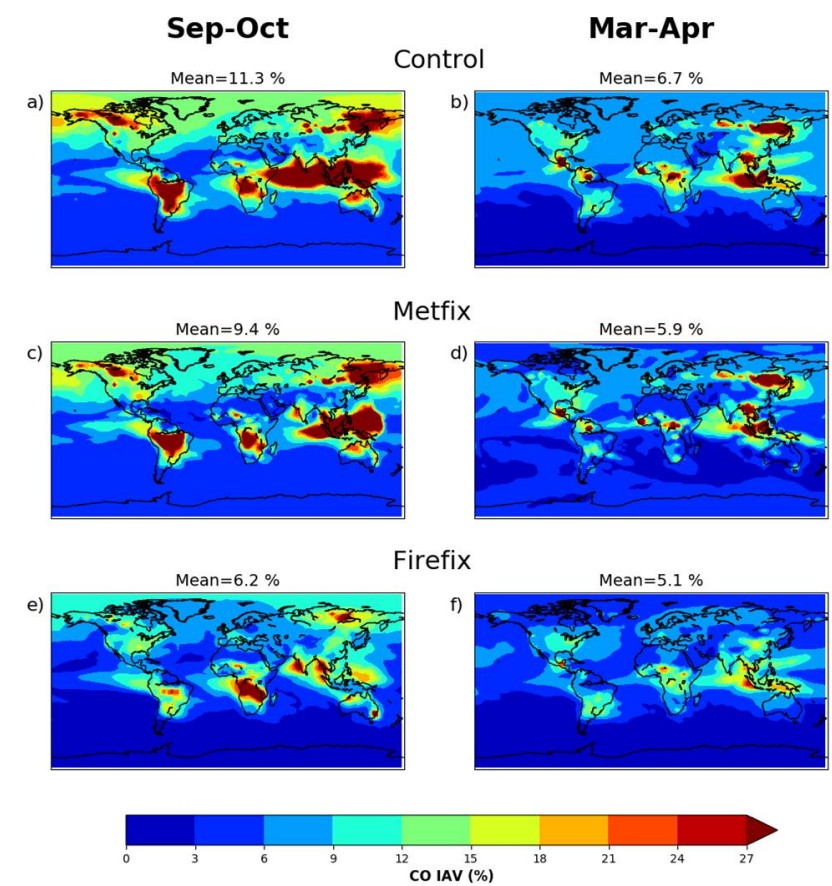

**Figure 5: The calculated interannual variability (coefficient of variation) of CO over the period 1997-2014 for September – October (left panels) and March - April (right panels) from (a, b) control simulation (CTRL), (c, d) fixed meteorology (METFIX) and (e, f) fixed fire emissions (FIREFIX).**

The IAV of OH and $O_3$ have more complex contributions from fire emissions and meteorology (Fig. 4b, c). For both species meteorology is the dominant cause of variability for the majority of the period, indicated by on-average greater deviation from CTRL in METFIX simulation than FIREFIX, including during El Niño events outside of the 1997 El Niño, such as in 2006. Our results compare well to Inness et al. (2015), who also found that changes to tropospheric $O_3$ during El Niño were driven by a combination of emissions and atmospheric dynamics. This is also in agreement with Doherty et al. (2006), where a strong correlation was found between ENSO meteorology and global $O_3$ burden, albeit with a lag period of several months. Various meteorological variables are known to affect OH and $O_3$ variability, including humidity, clouds and temperature (Stevenson et al., 2005; Holmes et al., 2013; Nicely et al., 2018). OH variability is particularly sensitive to changes in lightning $NO_x$ production which decreases during El Niño conditions (Turner et al., 2018). Murray et al. (2014) also examined factors affecting OH variability since the last glacial maximum, finding tropospheric water vapour, overhead stratospheric $O_3$ and lightning $NO_x$ to be key controlling factors. Furthermore, circulation changes during El Niño events have been linked to lower stratospheric $O_3$ variability (Zhang et al., 2015; Manatsa and Mukwada, 2017), which in turn influences tropospheric OH and $O_3$ concentrations (Holmes et al., 2013; Murray et al., 2014). Despite the importance of meteorological drivers, we find that fire emissions are the dominant cause of variation in both OH and $O_3$ during the 1997 El Niño, increasing global tropospheric $O_3$ burden by up to ~7% and decreasing tropospheric OH by up to ~6%. This result is supported by several other studies, which have found that during large fire events such as that caused by the 1997 El Niño, fire emissions substantially decrease tropospheric OH and increase tropospheric $O_3$ (Hauglustaine et al., 1999; Sudo and Takahashi, 2001; Holmes et al., 2013).

Our results indicate that while meteorology is generally the most important driver of IAV in global tropospheric OH and $O_3$, fire emissions can also play a key role and become the dominant driver when there are particularly large fire emissions related to El Niño.

Figure 6 shows the IAV of $O_3$, supporting the analysis of Fig. 4 that also suggests meteorology is the dominant process in controlling IAV. METFIX-simulated IAV differs substantially from the CTRL, with much lower IAV in Sept-Oct (33% decrease) and in Mar-Apr (42% decrease) when meteorology is repeated. However, in the METFIX run there remain peaks in variability in close proximity to regions with large biomass burning emissions, demonstrating the significant contribution from fire emissions. In the FIREFIX simulation the distribution of IAV is broadly similar to the CTRL simulation and with only a small change in global mean CV, indicating that fire emissions have less control on $O_3$ IAV. These results are again comparable to Voulgarakis et al. (2015) as the distribution of $O_3$ IAV in both CTRL and FIREFIX simulations is similar although with slightly larger values of variation due to differing time period.

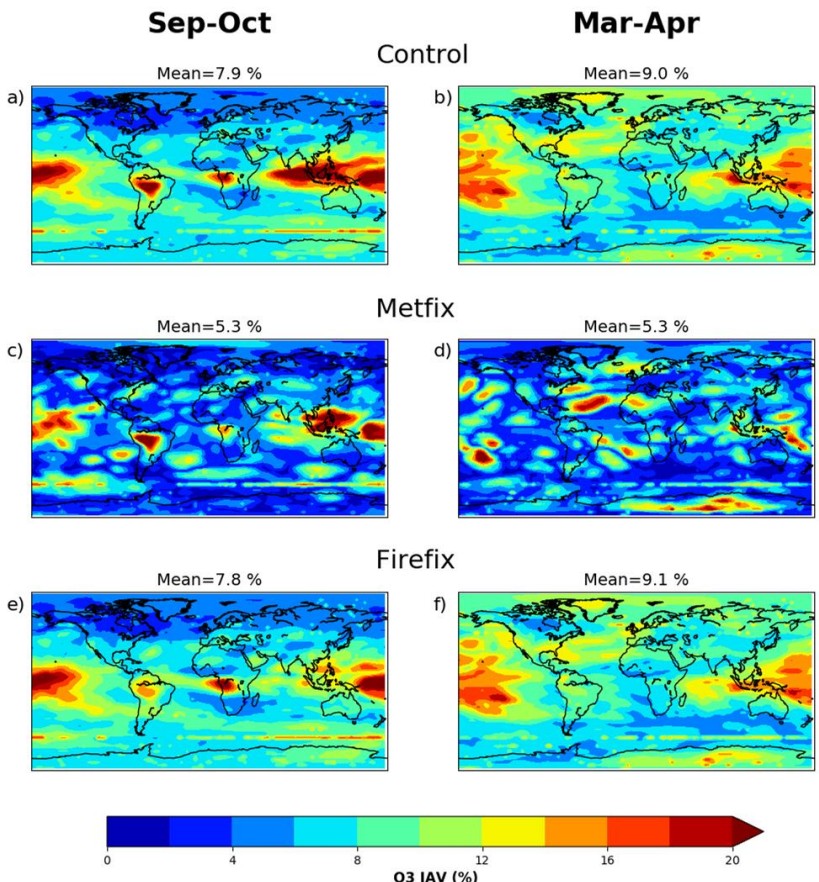

**Figure 6: The calculated interannual variability (coefficient of variation) of ozone over the period 1997-2014 for September – October (left panels) and March - April (right panels) from (a, b) control simulation (CTRL), (c, d) fixed meteorology (METFIX) and (e, f) fixed fire emissions (FIREFIX).**

**4.2 Indirect effect of CO on oxidation and lifetime of CH$_4$**

The COFIX sensitivity experiment was conducted to determine the indirect influence of CO emissions on $CH_4$ variability through changes in tropospheric OH concentrations. Figure 7a shows the difference in COFIX monthly mean OH concentrations from the control experiment, compared to that from the METFIX and FIREFIX simulations. When CO emissions from biomass burning are fixed, OH concentrations are consistently higher than in the CTRL simulation. This indicates that high CO emissions decrease global mean tropospheric OH. The greatest impact is during the 1997 El Niño where CO emissions were abnormally large, suppressing mass weighted global monthly mean OH concentrations by up to 9%. The mean effect on OH over the 1997 El Niño of -3.6% is comparable to that simulated by Butler et al. (2005), who also found an

increase in CO resulted in a change in OH of -2.2%. Duncan et al. (2003) found a similar magnitude response in OH to the Indonesian wildfires in 1997, of between -2.1% and -6.8%. The suppression of OH concentrations due to CO emission is also simulated to a lesser degree in the 2003 and 2006 El Niño events, but is absent in 2010 El Niño as this event had little impact on global fire occurrence (Randerson et al., 2017). The effect of fixing only CO from fires is greater than the effect of fixing all fire emissions due to co-emitted species such as $NO_x$, which act to increase OH concentrations.

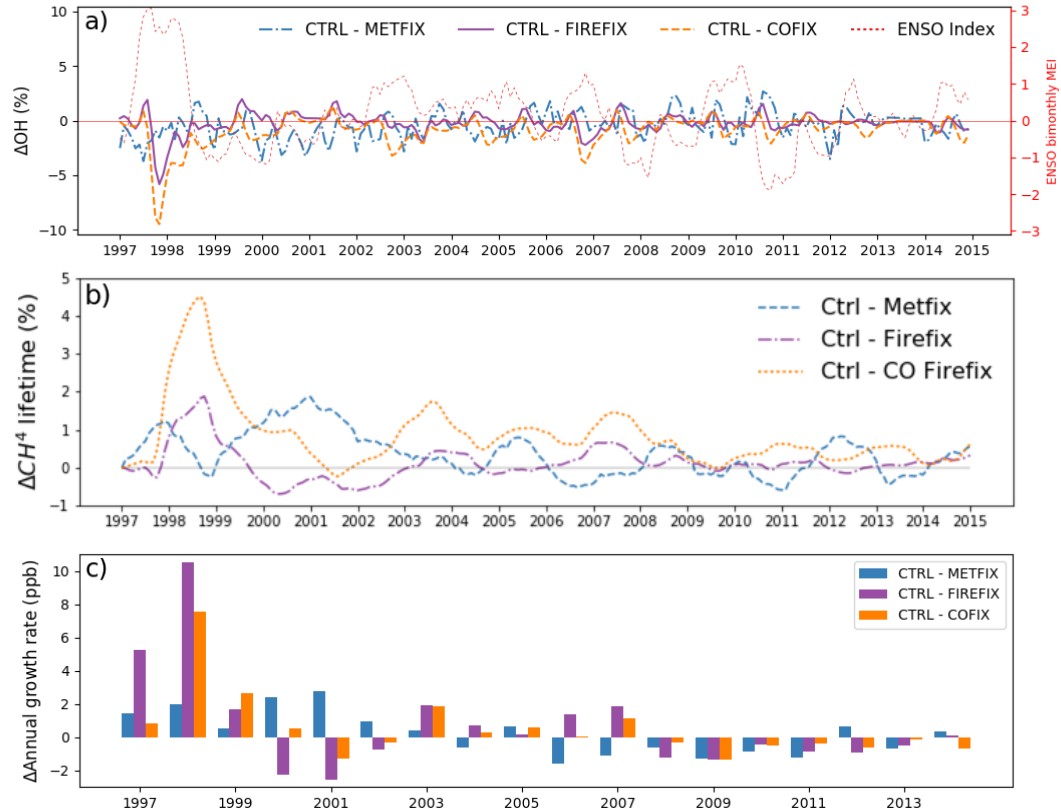

**Figure 7: Time series of (a) the change (%) in mass-weighted tropospheric OH, (b) change (%) in CH₄ lifetime and (c) resultant change (ppb) in annual CH₄ growth rate calculated using an offline box model. The ENSO bimonthly mean multivariate index is plotted in the dashed red line on the right-hand y-axis in panel (a).**

As OH is also the primary sink of $CH_4$ (~90%) (McNorton et al., 2016a), another effect of the decrease in OH due to CO emissions is to weaken the sink of $CH_4$, increasing its atmospheric lifetime. The magnitude of this can be seen in Fig. 7b; the COFIX simulation indicates that CO emissions from fires extend $CH_4$ atmospheric lifetime by more than 4% during the 1997 El Niño. Fixing all fire emissions also enhances $CH_4$ lifetime by around 2%. Increasing the lifetime of a species increases its concentration in steady-state equilibrium. Due to the scaling applied to $CH_4$ in TOMCAT we are unable to directly calculate the response in $CH_4$ growth rate from TOMCAT, as simulated global mean surface $CH_4$ concentrations are nudged to the observed value. Therefore, to determine the impact of the change to OH on $CH_4$ concentrations we used a simple global box model. This box model is similar to that described in McNorton et al. (2016a), which was found to compare well with other global and 12-box $CH_4$ models (Rigby et al., 2013; McNorton et al., 2016a). In this case, the box model used monthly mean tropospheric OH concentrations and $CH_4$ emissions for each simulation while assuming constant temperature to calculate the effect of changing OH on global mean surface $CH_4$. A fixed temperature was used as varying temperatures has been found to have a relatively small impact on derived $CH_4$ concentrations (McNorton et al., 2016a). The impact of fire emissions on the $CH_4$ growth rate is greatest in 1998, where all emissions from fires increased global $CH_4$ by 10.5 ppb (Fig. 7c). Analysis of the COFIX simulation demonstrates that up to 7.5 ppb (72%) of that change could have been caused by the release of CO alone and its role as a sink for OH. The effect on growth rate in the FIREFIX simulation is larger than the COFIX despite a

greater effect on $CH_4$ lifetime from the COFIX, due to directly emitted $CH_4$ varying with El Niño conditions in the COFIX simulation and not in FIREFIX. The influence of CO emissions on $CH_4$ growth rate calculated here is smaller than in Butler et al. (2005) despite a much larger effect on tropospheric OH. The radiative effect of the change to $CH_4$ from CO emitted from biomass burning alone in 1998 is 0.004 $Wm^{-2}$, calculated using updated expressions from Etminan et al. (2016).

## 4.3 Limiting factors of $O_3$ production

In this section we examine trends and the impact of El Niño on the production of tropospheric $O_3$. El Niño is known to have large effects on tropospheric $O_3$ precursors such as CO and $NO_x$, therefore examining $O_3$ production regimes during El Nino can provide insights into the main mechanism responsible for the observed changes in tropospheric $O_3$. The ratio between formaldehyde (HCHO) and nitrogen dioxide ($NO_2$) concentrations can be used to indicate the limiting factor for tropospheric $O_3$ production (Duncan et al., 2010). Ratios smaller than 1 indicate that removing VOCs will decrease tropospheric $O_3$ formation (i.e. a VOC-limited regime), while ratios larger than 2 indicate that removing $NO_x$ will reduce $O_3$ (i.e. a $NO_x$-limited regime). Ratios of 1-2 indicate that both $NO_x$ and VOC reductions could decrease $O_3$ (i.e. a 'both-limited' regime). Here we apply this methodology to determine the changes to this ratio from 1997-2014 and dependence of $O_3$ formation during the 1997 El Niño event. We compare the early period mean (1999-2003) to the end period mean (2010-2014) to determine whether significant changes have occurred over the 18-year period, and compared mean El Niño conditions to both.

In general, the SH and tropical regions have very high ratios, meaning they are strongly $NO_x$-limited (Fig. 8). The NH is also predominantly $NO_x$-limited although less robustly and polluted regions tend to be either VOC-limited or both-limited regimes. The ratio is largely constant across the modelled period, however there are some significant shifts such as in India, which was once solely $NO_x$-limited, becoming increasing VOC-limited due to increased $NO_x$ pollution (Hilboll et al., 2017). This shift in the spatial distribution of $O_3$ precursor emissions to lower latitudes leads to increased tropospheric $O_3$ production proportional to total emissions (Zhang et al., 2016).

During El Niño there are large changes, increasing the ratio and therefore $NO_x$ limitation by more than 40% in the Tropical Pacific. Significant changes to the ratio were also found in biomass burning regions of South America and Southeast Asia. This is due to the increase in $NO_x$ emissions in larger fire seasons associated with El Niño. However, these regions are already very heavily $NO_x$-limited due to high VOC emissions in forest regions, meaning that although the shift in HCHO/$NO_2$ ratio during El Niño is large, it is not substantial enough to alter the limiting factor for formation of tropospheric $O_3$ from one regime to another. Over India, El Niño conditions inhibit the trend towards a "both-limited" regime, as the $NO_x$-limited regime continues to dominate throughout.

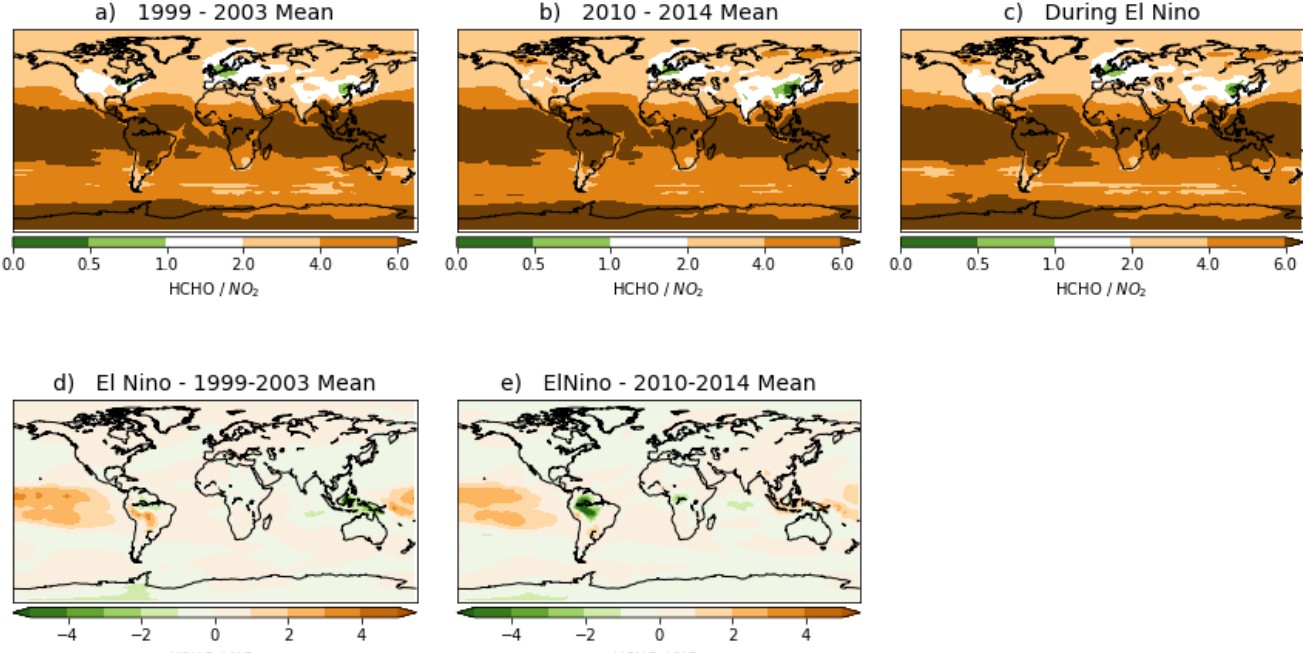

**Figure 8: Mean ratio of simulated tropospheric column HCHO to NO₂ amounts for (a) the beginning of model period (1999-2003), (b) the end of model period (2010-2014) and (c) during all El Niño events. Panels (d) and (e) show difference during El Niño from the 5-year mean values in panels (a) and (b), respectively.**

## 4.4 Impact on tropospheric ozone and radiative effects

The 1997 El Niño significantly altered the vertical distribution of $O_3$ in the troposphere, increasing $O_3$ concentrations in the NH while decreasing in the SH and tropics with an overall decrease in tropospheric $O_3$ of -0.82% compared to the 1997-2014 mean (Fig. 9a). In the CTRL simulation there is decreased $O_3$ in the tropical upper troposphere, possibly related to increased convection over the Eastern Pacific (Oman et al., 2013; Neu et al., 2014). We also simulate large increases in the mid-latitude upper troposphere of both hemispheres in the CTRL and FIREFIX simulations but not in METFIX, implying that this is

produced by El Niño-associated meteorological processes which promote intrusion of stratospheric air into the troposphere. These positive anomalies were also observed in Oman et al. (2013) and Zeng and Pyle (2005), attributed to El Niño influence on circulation patterns and enhanced stratospheric-troposphere exchange.

In general, the METFIX run simulates higher $O_3$ concentrations in the NH than the period mean and lower concentrations in the SH (Fig. 9b). This hemispherical shift is also present in the CTRL and FIREFIX simulations but with greater negative $O_3$

anomalies in the SH. The simulated NH increases in the CTRL simulation correspond to other studies of the 1997 El Niño (Koumoutsaris et al., 2008), while Oman et al. (2013) similarly reported negative $O_3$ anomalies in the SH during El Niño. Large increases in tropospheric $O_3$ in the Western Pacific, Indian Ocean and Europe contribute to the increase in $O_3$ in the NH, despite decreased $O_3$ in the Eastern Pacific (Chandra et al., 1998; Koumoutsaris et al., 2008; Oman et al., 2011).

There is an overall increase in $O_3$ (~2%) when meteorology was fixed to an ENSO-neutral year (i.e. 2013), meaning that

meteorology during the 1997 El Niño caused a decrease in tropospheric $O_3$ concentrations despite large increases in $O_3$ in regions of the upper troposphere due to stratospheric intrusion. During the 1997 El Niño we find a 0.4% increase in global tropospheric humidity compared to the period mean. This is likely partly responsible for the general decrease in $O_3$ due to meteorology, as increased humidity enhances $O_3$ loss (Stevenson et al., 2000; Isaksen et al., 2009; Kawase et al., 2011). Changes to transport and distribution of $O_3$ will also impact how efficiently tropospheric $O_3$ is produced and lost.

The similarities between the tropospheric O$_3$ distribution in the CTRL and FIREFIX simulations show that fire emissions have a relatively small impact on the global distribution of O$_3$, but do affect absolute values, as concentrations in the FIREFIX run are significantly lower at the tropics. This is likely because of the removal of large emissions of O$_3$ precursors in that latitude band when fire emissions are fixed to a non-El Niño year, as several studies have found that enhanced fires in 1997 El Niño increased tropospheric O$_3$ in the region (Chandra et al., 1998; Thompson et al., 2001; Doherty et al., 2006; Oman et al., 2013).


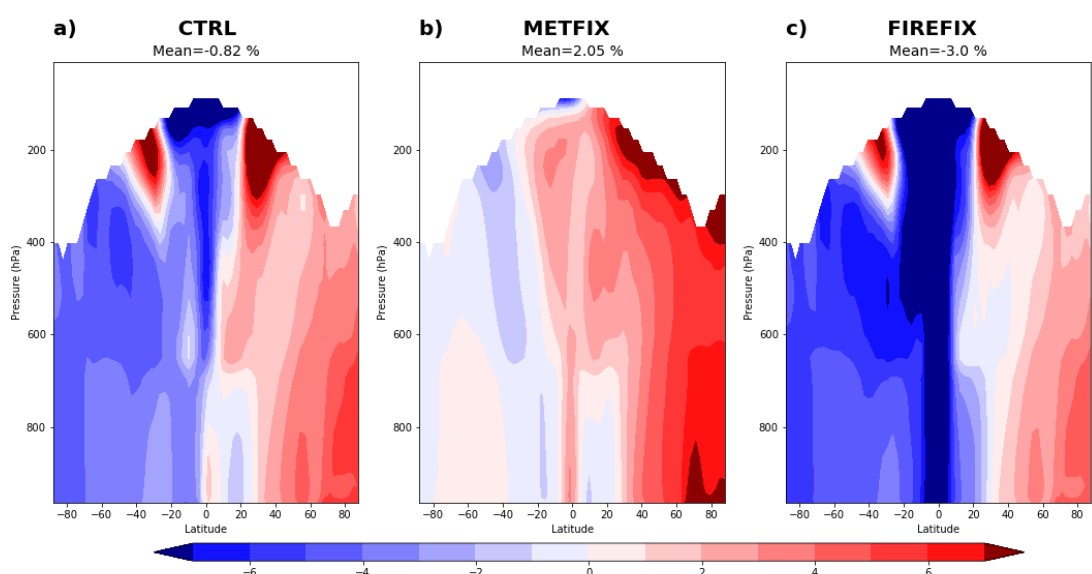

**Figure 9: Latitude-pressure cross sections of the percentage difference in O$_3$ concentrations during the 1997 El Niño event compared to 1997-2014 period mean for the TOMCAT simulations (a) CTRL, (b) METFIX and (c) FIREFIX simulations.**

Figure 10 shows the tropospheric O$_3$ radiative effect (RE) during the 1997 El Niño in each TOMCAT simulation, calculated using the Rap et al. (2015) tropospheric O$_3$ radiative kernel. Consistent with the relative changes in O$_3$ concentration, fire emissions and meteorology have contrasting effects on O$_3$ RE. When isolated, these effects are opposite and almost equal: fire emissions increase O$_3$ RE by 0.031 Wm$^{-2}$, while meteorology decreases by -0.030 Wm$^{-2}$. We performed an additional

simulation to determine the effect of these factors occurring simultaneously (BOTHFIX) and found the increasing effect from fire emissions to be dominant over the decreasing effect from meteorology, leading to an overall increase in global mean O$_3$ RE of 0.015 W m$^{-2}$. The effect of fire emissions occurs almost entirely over Indonesia and the Eastern Indian Ocean where the large influx of NO$_x$, CO and CH$_4$ from fire emissions during the 1997 El Niño causes large regional increases in tropospheric O$_3$. This increase, also observed in Chandra et al. (1998), causes a regional RE of up to 0.17 Wm$^{-2}$. Meteorology has more

varied impacts during El Niño, causing large decreases in O$_3$ RE over the Central Pacific Ocean (~ -0.36 Wm$^{-2}$) but also increases in the mid-latitudes of the Pacific Ocean (~0.33 Wm$^{-2}$). Globally the mean change to O$_3$ RE is small, around 0.015 Wm$^{-2}$, but large regional changes have the potential to significantly alter atmospheric heating and dynamics.

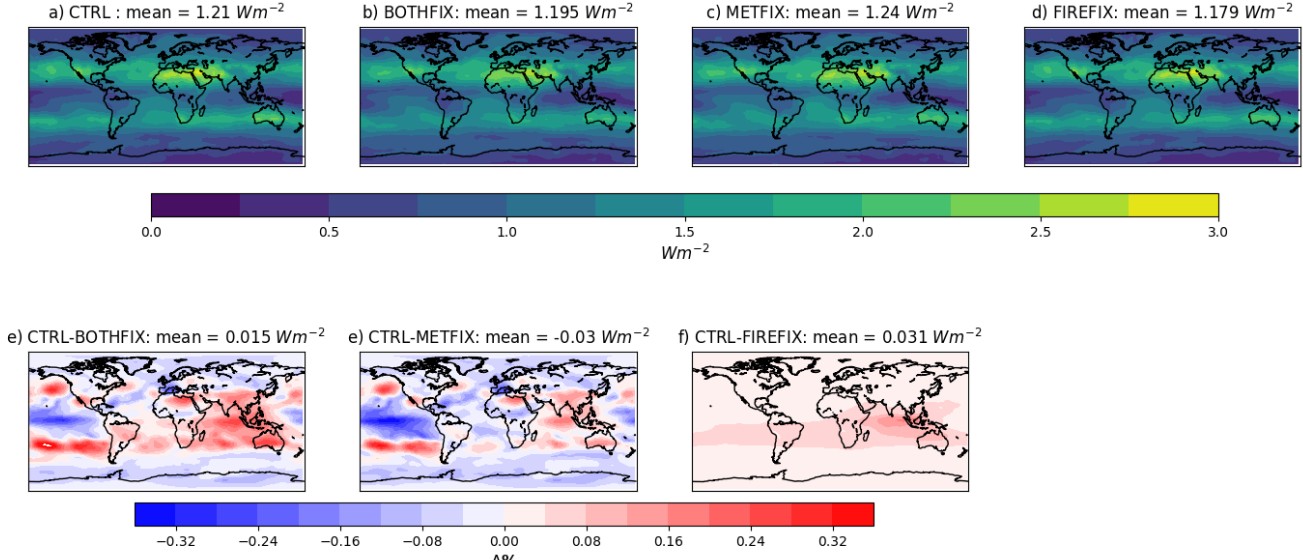

**Figure 10: Tropospheric O₃ radiative effects (Wm⁻²) from the TOMCAT simulations (a) control (CTRL), (b) fixed meteorology and fire emissions (BOTHFIX), (c) fixed meteorology only (METFIX) and (d) fixed fire emissions only (FIREFIX). Panels (e-g) show percentage differences between the control and the three perturbed simulations.**

## 6 Summary and conclusions

Global model simulations using annually invariant meteorology and fire emissions were performed for the period 1997-2014 in order to determine their relative impacts on the IAV of $O_3$ and $CH_4$, particularly during El Nino events. The TOMCAT-GLOMAP model used has been updated from that described by Monks et al. (2017) with improved cloud and photolysis representation and the introduction of Martensson sea spray emissions (Gordon et al., 2017). Model simulations were evaluated for a number of chemical species ($O_3$, $CH_4$, $NO_x$, CO) with observations from aircraft, satellites and ozone sondes. In general, the model shows a good agreement with observed values, although with some regional biases. Differences between model and observations may be due to a number of factors, such as the relatively coarse model resolution, uncertainties in the model emission inventories and errors in observations. However, good overall agreement of model simulations with different observations, including the ability of the model to simulate the observed atmospheric responses to El Niño events (i.e. OEI change of 2.8 DU compared to 2.4 DU in Ziemke et al. (2010)), provides confidence in model performance and results.

We find that the IAV of global CO concentrations is large and is primarily controlled by fire emissions over the modelled period. Exceptionally large CO emissions linked to El Niño in 1997 led to a decrease in OH concentrations of ~9%, which subsequently increased $CH_4$ lifetime by ~4%. Using a box model we quantify the isolated impact of this change in atmospheric chemistry on global $CH_4$ growth rate to be 7.75 ppb, ~75% of the total effect of fires. This effect, combined with concurrent direct $CH_4$ emission from fires, explains the observed changes to $CH_4$ growth rate during the 1997 El Niño.

Variability of oxidants $O_3$ and OH is far more dependent on meteorology than fire emissions, except during very large El Niño events, such as in 1997 and 1998, when fires become dominant in terms of total tropospheric burden, although meteorology still controls distribution. The change to tropospheric $O_3$ concentrations during El Niño has increased $O_3$ RE by 0.17 Wm⁻² over Southeast Asia and decreased by 0.36 Wm⁻² over the Central Pacific. The global mean $O_3$ RE change due to 1997 El Niño meteorology and fires is an increase of 0.015 Wm⁻², as emissions of $O_3$ precursors from fires causes increased $O_3$. El Niño also causes significant shifts in the ratio of HCHO:$NO_x$ – an indicator of $O_3$ production regime – but most significantly in the tropics which are heavily $NO_x$-limited, so this change does not cause a regime shift.

This work has shown that El Niño events significantly affect the variability of two important drivers of anthropogenic climate change. Further research into how El Niño events, with their associated effect on fire emissions, are likely to change in a warming climate is required to understand how these links between ENSO, $CH_4$ and $O_3$ may influence future climate change mitigation attempts.

**Author contribution**

MJR, AR and SRA conceptualised the study and planned the experiments. MPC, KJP, HG, WF and MJR developed and evaluated the version of TOMCAT-GLOMAP used here. BJK, BLL and RS provided the satellite retrievals for the $O_3$ comparison which was conducted by RJP. MPC and JM provided assistance and advice for the $CH_4$ box model. MJR performed the TOMCAT model runs, SOCRATES and box model calculations. MJR analysed the results with help from AR and SRA. MJR compiled results and prepared the manuscript. All co-authors contributed to the final version with comments.

**Acknowledgements**

M. Rowlinson is funded by a studentship for the NERC SPHERES Doctoral Training Partnership (NE/L002574/1). This work was undertaken using the ARCHER UK National Supercomputing Service (http://www.archer.ac.uk) and ARC3, part of the High Performance Computing facilities at the University of Leeds, UK.

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
