# Peer review of "Impact of El Niño Southern Oscillation on the interannual variability of methane and tropospheric ozone"

_Atmospheric Chemistry and Physics, 2019_

## Referee Comment (RC1) · Anonymous Referee #1 · 19 Apr 2019

In this paper, the authors document the results of model experiments analyzing the influence of ENSO on interannual variability of carbon monoxide, tropospheric ozone, hydroxyl radical and resulting impact on radiative effects and methane variability. The paper is generally well-organized and addresses scientific questions within the scope of ACP. My two main concerns are: a) the ability of the model to capture observations during ENSO years has not been explicitly demonstrated and b) results from previous studies (both model and observational) on the influence of ENSO on tropospheric ozone have not been considered (though studies on IAV of CO have been considered). These issues are highlighted below with specific suggestions on further improving the manuscript.

[Figure]

Specific Comments:

Abstract: To me, the first sentence of the abstract gives the impression that the focus of this study is the analysis of methane trends and variability which is currently being intensely debated in the literature (Turner et al., 2019; Nisbet et al., 2019). Unless I misunderstood, the paper is geared towards analyzing the impact of ENSO on carbon monoxide, tropospheric ozone, hydroxyl radical as well as methane. The influence of ENSO on methane variability is discussed here but a full analysis of the methane growth rate is not performed in this study. Lines 45-47 better reflect the analyses performed here. The first couple of sentences should be revised so that they accurately represent the focus of this study which I understand to be ENSO driven changes in atmospheric composition and their impacts rather than just methane growth rate.

Section 2.1: How does the model calculate biogenic VOC emissions? Given that VOC oxidation is an important source of CO (about 15% according to Duncan et al. 2007), I would imagine that variability in VOC emissions (driven by variations in meteorology, radiation, land-use, CO2) (Lathiere et al., 2005) would have some impact on CO IAV.

L133-134: Emissions of "all source of methane have been included in the model." Could you please elaborate on which sources of methane emissions have been included in the model?

L135: Replace nitrous oxide (which is N2O) with nitrogen oxide.

L137-139: What emissions does JULES simulate? Wetlands? Agriculture? Please clarify.

Section 3: Given that the model is being used to analyse the impact of El Nino, in addition to the climatological evaluation discussed in this section, a more focused evaluation against measurements in El Nino years would more appropriately build confidence in the model's ability to capture features unique to conditions in these years.

L167-168: Are averaging kernels applied to model output for evaluation against satellite

observations. Please clarify for MOPITT CO and OMI ozone.

L190-191: It is stated on L139-140 that surface methane concentrations are scaled in the model to observed values. It is therefore not surprising that the model performs well near the surface for methane. I think this sentence should be caveated.

L191-192: It looks like the model O3 is a factor or two too low compared with aircraft observations over southern Africa and off the coast over southern Atlantic. Please elaborate on the possible reasons for this bias.

L195-196: It would be helpful to have a quantitative estimate (e.g., bias, error, correlation) of how well the model captures the observations.

L196-197: By how much are the simulated concentrations of NOx lower than the observations? What processes (emissions, chemistry or meteorology) are likely responsible for these biases? Do these biases in NOx have implications for the simulation of ozone?

Section 3.2: It would be useful to clarify that the OH evaluation is performed for year 2000. How is the tropopause determined to calculate tropospheric OH concentrations? How does the model tropospheric methane lifetime (due to OH reaction only) compare against that derived from observational estimates by Prather et al. (2012)?

L215-216: What caused the lowering of OH in this version of TOMCAT versus that described by Monks et al (2017)?

L218-219: As I understand, the authors choose to calculate the chemical lifetime rather than the atmospheric lifetime of methane from the model because not all loss processes affecting the atmospheric lifetime are considered in the model (e.g., soil uptake, stratospheric loss, tropospheric loss due to chlorine). And not "Due to the long lifetime of CH4". Please rephrase this sentence.

L246-247: I think this sentence should be placed before describing the Voulgarakis et al (2015) results.

L250-252: Is the simulated IAV in tropospheric CO concentrations driven by biomass burning emissions similar to the IAV in the imposed biomass burning emissions? I would imagine that the IAV in GFED4 CO emissions would be similar to that for CO simulated by the model. Would be useful to confirm this. This then begs the question - what is driving the interannual variability in biomass burning emissions - is it changes in area burnt, biomass available for burning, or meteorological conditions or all of these?

L297-298: I find this sentence confusing - is it that including CO results in a decreasing trend in OH? Though, this is not evident from figure 6.

L316-317: Given that the reaction rate constant of CH4+OH is strongly sensitive to temperature, approximately 2 % K $-1$ (John et al., 2012), what is the impact of assuming constant temperature in the box model on the results discussed here?

L332-L333: The authors mention that they compared the early mean period (1997-2001) with the end period but Figure 7 shows the early period as 1999-2003 without the influence of El Nino. Please revise this sentence.

L335: Remove "a".

L337-338: The influence of shift in ozone precursor emissions from the mid-latitudes to the tropics has been demonstrated by Zhang et al. (2016).

L344: This sentence is confusing. From figure 7b, it looks like India becomes "both-limited" towards the end of the simulation.

L347-349: Do observations also indicate this significant alteration of the vertical distribution of tropospheric ozone? How does this interpretation of the model results compare with the analysis of Chandra et al (1998) and Ziemke and Chandra (1999)?

L385-L387: These are broad-brush statements which then make it easy to question the model's ability to simulate chemical composition during El Nino years and derived interpretations. As I mentioned above, a more focused evaluation would be helpful to reveal model strengths and weaknesses building confidence in the results.

References:

Duncan et al. 2007, https://agupubs.onlinelibrary.wiley.com/doi/10.1029/2007JD008459. Chandra et al. 1998, https://agupubs.onlinelibrary.wiley.com/doi/10.1029/98GL02695. John et al., 2012, https://www.atmos-chem-phys.net/12/12021/2012/. Lathiere et al., 2006, https://www.atmos-chem-phys.net/6/2129/2006/acp-6-2129-2006.html. Nisbet et al., 2019, https://agupubs.onlinelibrary.wiley.com/doi/10.1029/2018GB006009. Turner et al., 2019, https://www.pnas.org/content/116/8/2805 Zhang et al., 2016, https://www.nature.com/articles/ngeo2827. Ziemke and Chandra, 1999, https://agupubs.onlinelibrary.wiley.com/doi/abs/10.1029/1999JD900277.

---

## Referee Comment (RC2) · Anonymous Referee #2 · 23 Apr 2019

General comment:

The manuscript "Impact of El Niño Southern Oscillation on the interannual variability of methane and tropospheric ozone" written by Matthew J. Rowlinson describes the impact of meteorological variability and forest fires associated with El Niño Southern Oscillation (ENSO) on methane and tropospheric ozone. The manuscript contains novel investigation to quantitatively isolate the impact of forest fire emissions and meteorological variability on methane lifetime and growth rate using modeling approach. While many modeling studies on the impact of ENSO on tropospheric ozone have been conducted, this is an interesting work that deduced spatial variations in radiative effects of

tropospheric ozone changes during El Niño. The authors used appropriate model simulations to use for this science problem. Overall this manuscript is well written and easy to follow. I would like to consider the publication of this manuscript from Atmospheric Chemistry and Physics after minor revision. Please see the following comments.

Specific comments:

1. Model evaluation

In Section 3, the authors presented the model evaluation of mean concentration fields of O3, CO, CH4, and NOx with satellite, aircraft, and ozonesonde observations, whereas the authors do not conduct model evaluation of inter-annual variations in these concentration fields during El Niño. The model evaluation would also be helpful to support validity of the model simulations used in this study.

2. Impact of ENSO on OH and methane

In Section 4.2, the authors quantified the impact of forest fire emissions on CH4 growth rate; however, the authors do not compare it with the observed CH4 growth rate, even though the authors concluded that "This effect, combined with concurrent direct CH4 emission from fires explain the observed changes to CH4 growth rate during the 1997 El Niño" (P. 15, L. 391—392) in Section 6. I would like to recommend to add the comparison of the observed and simulated CH4 growth rate in Section 4.2.

The authors also presented the impacts of forest fire emissions and meteorological variability on OH concentrations in Figure 6. Although the impacts are comparable except during the 1997—1998 El Niño, the authors hardly mentioned the impact of meteorology. A more detailed analysis (e.g., how do specific humidity, cloud, and lightning NOx affect inter-annual variations in OH?) would also be interesting, though this may be beyond the scope of the current manuscript.

P. 2, L. 57: Tropospheric O3 changes related to meteolorogy were attributed by both transport and O3 loss with water vapor in Section 4.3; Why do the authors mention

only atmospheric transport changes in abstract?

P. 4, L. 139—140: Why is CH4 concentrations scaled to the observations even though the CH4 emission inventory is used in the model?

P. 8, L. 242—243: What is definition of El Niño periods used in this study? Please clarify the definition.

P. 8, L. 253—255: I would like to recommend to add the percentage number of the increase during the 2002—2003, 2006, and 2009—2010 El Niño events to compare them with the extreme El Niño event in 1997—1998 quantitatively.

P. 9, L. 271—272: Why does fire emissions have small impact on CO IAV?

P. 11, L. 300—302: Do you have possible explanation of the difference between this study and Butler et al. (2005)? Did you compare increases in CO emissions in GFED4 with Butler et al. (2005)?

P. 13, L. 327: Please clarify the reason why you conducted this analysis in Section 4.3.

P. 14, L. 346—362: Many previous works have been done with different models and satellite observations with regard to ENSO impacts on tropospheric ozone. It would be beneficial to see more discussion of how the results presented here compare to previous studies (e.g., Stevenson et al., 2005; Zeng and Pyle, 2005; Doherty et al., 2006; Koumoutsaris et al., 2008; Nassar et al., 2009; Ziemke et al., 2010; Sekiya and Sudo, 2012; Oman et al., 2013; Neu et al., 2014; Inness et al., 2015).

P. 14, L. 364—365: Do the authors used TES O3 radiative kernel? Please clarify what data the authors used in this study.

Technical corrections:

P. 2. L. 58: typo for nitrogen oxides?

P. 4, L. 135: typo for nitrogen oxides?

P. 13, L. 332: Which is correct, "1997-2001" or "1999-2003"? The "1999-2003" period would be appropriate to obtain the mean fields.

---

## Author Comment (AC1) · 5 Jun 2019

Response to reviewer comments for the manuscript: **Impact of El Niño Southern Oscillation on the interannual variability of methane and tropospheric ozone** by **Rowlinson et al.**

We thank the two reviewers for their detailed feedback on our manuscript. We have now carefully revised the manuscript according to all the comments provided. To guide the review process we have copied the reviewer comments below (in black) and provided our responses (in blue).

**Responses to reviewer #1:**

**Reviewer Summary:**
In this paper, the authors document the results of model experiments analyzing the influence of ENSO on interannual variability of carbon monoxide, tropospheric ozone, hydroxyl radical and resulting impact on radiative effects and methane variability. The paper is generally well-organized and addresses scientific questions within the scope of ACP. My two main concerns are: a) the ability of the model to capture observations during ENSO years has not been explicitly demonstrated and b) results from previous studies (both model and observational) on the influence of ENSO on tropospheric ozone have not been considered (though studies on IAV of CO have been considered). These issues are highlighted below with specific suggestions on further improving the manuscript.

**Authors' response:** We would like to thank the reviewer for the positive and constructive comments on our manuscript. We agree that the two major concerns raised are important and improve the manuscript. We have now substantially revised our study to address both major concerns. In particular:
   a) We conducted further evaluation of the model in order to demonstrate its ability to capture ENSO events. This has been done using the Ozone ENSO Index (OEI) proposed by Ziemke et al. (2010) and by comparing regional tropospheric $O_3$ changes during El Niño events to previous studies. These developments are explained in detail below.
   b) We included considerably more discussion of the influence of El Niño on tropospheric $O_3$, detailing how our results compare with previous literature and providing explanations for the responses simulated in our model.

**Specific Comments:**
Abstract: To me, the first sentence of the abstract gives the impression that the focus of this study is the analysis of methane trends and variability which is currently being intensely debated in the literature (Turner et al., 2019; Nisbet et al., 2019). Unless I misunderstood, the paper is geared towards analysing the impact of ENSO on carbon monoxide, tropospheric ozone, hydroxyl radical as well as methane. The influence of ENSO on methane variability is discussed here but a full analysis of the methane growth rate is not performed in this study. Lines 45-47 better reflect the analyses performed here. The first couple of sentences should be revised so that they accurately represent the focus of this study which I understand to be ENSO driven changes in atmospheric composition and their impacts rather than just methane growth rate.

**Authors' response:** We thank the reviewer for this point and we agree that the abstract needs to be reformulated to avoid any confusion and to better summarise the findings of the paper.
We have now revised the abstract to more accurately convey the scope of the paper:

**Changes in manuscript:**
**L43-46**

"The interannual variability of greenhouse gases methane ($CH_4$) and tropospheric ozone ($O_3$) is largely driven by natural variations in global emissions and meteorology. The El Niño Southern Oscillation (ENSO) is known to influence fire occurrence, wetland emission and atmospheric circulation, affecting sources and sinks of $CH_4$ and tropospheric $O_3$, but there are still important uncertainties associated with the exact mechanism and magnitude of this effect."

**Section 2.1:** How does the model calculate biogenic VOC emissions? Given that VOC oxidation is an important source of CO (about 15% according to Duncan et al. 2007), I would imagine that variability in VOC emissions (driven by variations in meteorology, radiation, land-use, CO2) (Lathiere et al., 2005) would have some impact on CO IAV.

**Authors' response:** In our study, biogenic VOC emissions are not calculated in the model but read in from the MEGAN-MACC biogenic emissions inventory. The emissions are fixed-year so we do not simulate the effect of BVOC IAV on CO IAV, however monthly and seasonal variability is accounted for. We have now clarified this in the text.

**Changes in manuscript:**
**L141-142**
"Monthly varying biogenic VOC emissions are from the MEGAN-MACC emissions inventory for reference year 2000, calculated from the Model of Emissions of Gases and Aerosols from Nature (MEGANv2) (Sindelarova et al., 2014)."

We now discuss this in Section 4.1 as a possible explanation for the slightly lower CO IAV calculated compared to Voulgarakis et al. (2015), however although the seasonal variability of BVOC emissions is high (17-25%), studies have found the interannual variation to be relatively minor (2-4%) (Naik et al., 2004; Lathière et al., 2005). Therefore, although we acknowledge that we do not account for CO IAV from BVOC IAV, the impact is likely small. In the text we now discuss the importance of BVOC oxidation as a source of CO and the relatively small interannual variability of BVOC emissions.

**Changes in manuscript:**
**L296-299**
"The slightly lower estimate here may be a result of the fixed-year BVOC emissions, removing the effect of IAV of biogenic emissions on CO IAV. BVOC oxidation is estimated to contribute 15% of the total source of CO (Duncan et al., 2007), however the IAV of BVOC emissions has been found to be relatively small, ~2-4% (Naik et al., 2004; Lathière et al., 2005)."

L133-134: Emissions of "all source of methane have been included in the model." Could you please elaborate on which sources of methane emissions have been included in the model?

**Authors' response:** We agree it is important to be more specific here, so we have expanded on this statement in the text.

**Changes in manuscript:**
**L142-146**
"The $CH_4$ inventory was produced by (McNorton et al., 2016b), with wetland emissions derived from the Joint UK Land Environment Simulator (JULES) and biomass burning emissions from GFEDv4 (Randerson et al., 2017). These are then combined with anthropogenic emissions from EDGARv3.2, paddy field emissions from Yan et al. (2009) and termite, wild animal, mud volcano, hydrate and ocean emissions from Matthews and Fung (1987) (McNorton et al., 2016b)."

L135: Replace nitrous oxide (which is N2O) with nitrogen oxide.

**Authors' response:** Thank you for pointing out the mistake. This has now been corrected.

L137-139: What emissions does JULES simulate? Wetlands? Agriculture? Please clarify.

**Authors' response:** This has now been addressed in the reply to a previous comment and the addition of new text at **L142-146**.

Section 3: Given that the model is being used to analyse the impact of El Nino, in addition to the climatological evaluation discussed in this section, a more focused evaluation against measurements in El Nino years would more appropriately build confidence in the model's ability to capture features unique to conditions in these years.

**Authors' response:** We agree that this is an important point that would add confidence in the model and our conclusions. We now clarify in section 2.1 that our model is driven by ECMWF reanalysis, which has previously been shown to be able to represent ENSO events.

**Changes in manuscript:**
**L133-135**
"ECMWF ERA-Interim reanalyses have been shown to have good skill in capturing Madden-Julian Oscillation (MJO) events which in turn impact the onset of ENSO events (Dee et al., 2011), giving confidence that the model competently represents El Niño meteorological conditions."

We have now also compared simulated tropospheric $O_3$ response to El Niño events against observed responses in literature. We calculate an Ozone ENSO Index (OEI) based on Ziemke et al. (2010), which yields a response of +2.8 DU in the model. This compares with a +2.4 DU response in observations given in Ziemke et al. (2010). We have also added a new figure in the supplementary material showing regional response in total $O_3$ column and compared this to findings of Zhang et al. (2015) (Fig. S5).

**Changes in manuscript:**
**L197-204**
"We have also assessed the capability of TOMCAT-GLOMAP to simulate observed responses to El Niño events. Ziemke et al. (2010) derived an $O_3$ ENSO index using satellite observations, finding that for a +1K change in the Nino 3.4 index, there was a 2.4 DU increase in the OEI. In TOMCAT-GLOMAP, we calculate a 2.8 DU increase per +1K in the Nino 3.4, indicating a slightly larger but comparable response to El Niño events. The regional response of tropospheric $O_3$ to El Niño was evaluated against an analysis using various observations and a chemistry-climate model in Zhang et al. (2015). That study observed increased total $O_3$ column in the North Pacific, southern USA, north-eastern Africa and East Asia, with decreases over central Europe and the North Atlantic. All of these observed responses were present in TOMCAT-GLOMAP simulations, except with a slight increase in TOC in central Europe and a simulated decrease in Western Europe and East Atlantic (Fig. S5). "

L167-168: Are averaging kernels applied to model output for evaluation against satellite observations? Please clarify for MOPITT CO and OMI ozone.

**Authors' response:** Averaging kernels were applied to the model output before they could be compared against satellite retrievals. This is now clarified in detail in the supplementary, with an explanatory sentence added in section 3.

**Changes in manuscript:**
**L176-178**
"MOPITT satellite retrievals have been used to evaluate CO at 800 hPa and 500 hPa (Emmons et al., 2004) and are shown in Fig. S1 and S2, respectively, along with a description of satellite product and averaging kernels applied to the model output."

L190-191: It is stated on L139-140 that surface methane concentrations are scaled in the model to observed values. It is therefore not surprising that the model performs well near the surface for methane. I think this sentence should be caveated.

**Authors' response:** We thank the reviewer for this observation as this is an important point which was not made clear in the original manuscript. The scaling in TOMCAT-GLOMAP affects only the simulated global mean surface methane concentration, scaling this value to the observed global mean surface concentration. The spatial distribution and vertical transport of methane is still simulated and therefore relevant to be evaluated. The description of the methane scaling has been revised:

**Changes in manuscript:**
**L146-148**
"The global mean surface $CH_4$ mixing ratio is scaled in TOMCAT-GLOMAP to a best-estimate based on observed global surface $CH_4$ mean concentration (McNorton et al., 2016a; Dlugokencky, 2019)."

However, we agree that the statement on model performance for methane simulation should be caveated. We have revised the statement now on L215-217:

**Changes in manuscript:**
**L219-222**
"Absolute concentrations of $CH_4$ in TOMCAT simulations match aircraft data very well, although given the global mean surface concentration scaling we expect the magnitude of $CH_4$ to be well simulated. The latitudinal and vertical distributions are also well captured, giving confidence in the model transport and OH simulation."

L191-192: It looks like the model $O_3$ is a factor or two too low compared with aircraft observations over southern Africa and off the coast over southern Atlantic. Please elaborate on the possible reasons for this bias.

**Author response:** We thank the reviewer for highlighting this. Following this comment we have reassessed the comparison of simulated $O_3$ with the aircraft data, and extended the evaluation of $O_3$ using ozone sondes (Fig 1). The limited temporal coverage of the aircraft data means that we were comparing long-term mean simulated concentrations with patchy observational data. While for other species (CO, CH4 and PAN) this is the only option to carry out basic evaluation of the model and compare broad characteristics, for $O_3$ the OMI satellite retrievals and ozone sonde climatologies from Tilmes et al. (2012) provide a much better, more fair evaluation. The ozone sonde climatologies are long-term mean concentrations over a period directly comparable to our observations (1995-2011), therefore offering a more reliable comparison.
We therefore decided to remove the $O_3$ comparison with the aircraft data and conduct instead a more thorough evaluation of tropospheric $O_3$ against the ozone sonde climatology. We have now included a new figure in the manuscript (Fig. 1), illustrating this comparison. We believe the $O_3$ evaluation with both ozone sondes and satellite data provides a good

indication of the model's skill at simulating tropospheric O$_3$. For the other species where we present no alternative comparisons, we retain the aircraft comparisons (now Fig. 2), but have pointed out explicitly the caveats associated with these comparisons.

**Changes in manuscript:**
**L208-2111**
"While the comparison of observational data from intermittent aircraft campaigns does not offer a perfect comparison with the model simulated long-term mean concentrations, it allows evaluation of broad characteristics of a number of species over vertical profiles in many global regions."

L195-196: It would be helpful to have a quantitative estimate (e.g., bias, error, correlation) of how well the model captures the observations.

**Authors' response:** We agree that is important to be more quantitative here. We have now calculated normalised mean biases of simulated tropospheric O$_3$ against ozone sonde observations from Tilmes et al. (2012). This is now included in the text as Figure 1, with discussion of the calculated bias at the beginning of section 3.

**Changes in manuscript:**
**L190-194**
"TOMCAT O$_3$ has also been evaluated using sonde observations (Fig. 1 and S4) (Tilmes et al., 2012), with TOMCAT generally representing the vertical profiles, seasonal variation and absolute concentrations of O$_3$ very well, with a normalised mean bias (NMB) of 1.1% across all sites at 700-1000 hPa and 2.1% at 300-700 hPa. The model capably simulates seasonality of tropospheric O$_3$ (Fig. 1), with a maximum seasonal bias of 6.3% at 300-700 hPa in March-May."

We have also calculated normalised mean biases against the aircraft observations for CH$_4$, CO and PAN. This figure is now included in the supplementary material (Fig. S6). We have also mentioned specific bias values in the discussion in section 3.1.

**Changes in manuscript:**
**L217-227**
CO concentrations decrease with altitude but the largest values still occur around urban areas and burning regions, which can be seen in both model and aircraft concentrations. Consistent with the comparison with MOPITT satellite retrievals (Fig. S1 and S2), the model underestimates CO concentrations particularly near the surface, with a NMB of -11.1%, -9.93% and -0.25% at 0-2 km, 2-6 km and 6-10 km, respectively. Absolute concentrations of CH$_4$ in TOMCAT simulations match aircraft data very well, although given the global mean surface concentration scaling we expect the magnitude of CH$_4$ to be well simulated. The latitudinal and vertical distributions are also well captured, giving confidence in the model transport and OH simulation. Aircraft observations show CH$_4$ also decreases with altitude and the hemispherical disparity becomes more pronounced, with higher concentrations in the NH. For PAN concentrations, the simulated spatial distribution is broadly well captured, as is the increased concentration with altitude. There is a general low bias in absolute concentrations near the surface (NMB=-12.3%), with better comparison at 2-6 km (NMB=1.68%) and over-estimation at 6-10 km (NMB=18.17%)."

L196-197: By how much are the simulated concentrations of NOx lower than the observations? What processes (emissions, chemistry or meteorology) are likely responsible for these biases? Do these biases in NOx have implications for the simulation of ozone?

**Authors' response:** The low NO$_x$ values we simulate in TOMCAT relative to observations are very likely a result of the lower temporal and spatial resolution of the model output

compared with the spatial scale on which strong gradients in $NO_x$ are observed. The short atmospheric lifetime of $NO_x$ makes it difficult to compare aircraft measurements to modelled values (Huijnen et al., 2010). Underestimation of $NO_x$ concentrations would affect tropospheric $O_3$, decreasing $O_3$ concentrations in non-urban locations and potentially causing the slight low bias in $O_3$ in the model. However, in this study we have conducted a thorough evaluation of tropospheric $O_3$ which gives us confidence that the model capably simulates $O_3$. We therefore decided that a direct evaluation of $NO_x$ in TOMCAT did not add much to our analysis and it was removed from the main manuscript and replaced with an evaluation for peroxyacetyl nitrate (PAN) (Fig 2j-l). The relatively long-lived lifetime of PAN means it is a reservoir species with a less heterogeneous distribution, therefore this comparison gives a better indication of model transport and chemistry performance (Huijnen et al., 2010).

**Changes in manuscript:**
**L223-227**
"For PAN concentrations, the simulated spatial distribution is broadly well captured, as is the increased concentration with altitude. There is a general low bias in absolute concentrations near the surface (NMB=-12.3%), with better comparison at 2-6 km (NMB=1.68%) and over-estimation at 6-10 km (NMB=18.17%). "

Section 3.2: It would be useful to clarify that the OH evaluation is performed for year 2000. How is the tropopause determined to calculate tropospheric OH concentrations? How does the model tropospheric methane lifetime (due to OH reaction only) compare against that derived from observational estimates by Prather et al. (2012)?

**Authors' response:** The tropopause in our study was a climatological tropopause calculated from the method in Lawrence et al. (2001), using the formula:

$$\rho_{cli} = 300 - 215(\cos(\phi))^2$$

Where $\phi$ is latitude and $\rho$ is pressure. We have revised the text to clarify this and also mentioned the year which was used for the evaluation:

**Changes in manuscript:**
**L230-232**
"Here we follow the evaluation methodology recommended by Lawrence et al. (2001) of dividing tropospheric OH into 12 sub-domains, from the surface to a climatologically derived tropopause. This method was also used to evaluate a previous version of TOMCAT(vn1.76) by Monks et al. (2017) allowing direct comparison. The evaluation is performed for the year 2000."

The tropospheric methane lifetime in Prather et al. (2012) is 11.2 ± 1.3 years. The lifetime from TOMCAT is below this range, however the lifetime calculated here is comparable to other model estimates in Naik et al. (2013). In addition, the distribution of OH in TOMCAT can explain the slightly lower lifetime. Mean tropospheric OH in TOMCAT fits well with other estimates (Prinn et al., 2001; Wang et al., 2008), but the OH evaluation in section 3.2 indicates that in TOMCAT OH concentrations are larger in the lower troposphere than other estimates, especially in the NH. This is also the region where methane concentrations are highest, thereby increasing the sink and decreasing the chemical methane lifetime in TOMCAT relative to other studies.

L215-216: What caused the lowering of OH in this version of TOMCAT versus that described by Monks et al (2017)?

**Authors' response:** The main development of TOMCAT-GLOMAP since Monks et al. (2017) is the introduction of improved cloud fields based on reanalyses, replacing the previously used climatologies. This leads to photolysis rate changes which then affect OH concentrations. We now add the following in the manuscript:

**Changes in manuscript:**
**L244-246**
"This is primarily due to an updated treatment of clouds, in which climatological cloud fields have been replaced with cloud fraction from ECMWF reanalyses data, affecting photolysis rates."

L218-219: As I understand, the authors choose to calculate the chemical lifetime rather than the atmospheric lifetime of methane from the model because not all loss processes affecting the atmospheric lifetime are considered in the model (e.g., soil uptake, stratospheric loss, tropospheric loss due to chlorine). And not "Due to the long lifetime of CH4". Please rephrase this sentence.

**Authors' response:** We agree that this point needs clarification. The reason for the calculation of the chemical lifetime was a combination of simplified treatment of methane in the model (i.e. the scaling) and its relatively long lifetime. Furthermore, given the focus of the paper on OH and atmospheric chemistry changes, the chemical lifetime was the most relevant measure. We have now amended the text as follows:

**Changes in manuscript:**
**L247-250**
"Due to the simplified treatment of $CH_4$, the scaling applied and its relatively long atmospheric lifetime, the total atmospheric lifetime cannot be determined from TOMCAT simulations. Instead a chemical lifetime due to reaction with OH is calculated from $CH_4$ and OH burdens, disregarding stratospheric and soil sinks (Fuglestvedt et al., 1999; Berntsen et al., 2005; Voulgarakis et al., 2013)."

L246-247: I think this sentence should be placed before describing the Voulgarakis et al (2015) results.

**Authors' response**: Thank you for this suggestion. This has now been reworded and moved as suggested.

**Changes in manuscript:**
**L281-284**
"Conversely, a study by Monks et al. (2012) considered CO IAV in the Arctic, finding that biomass burning was the dominant driver with a strong correlation to El Niño. Voulgarakis et al. (2015) also suggested that biomass burning was the more important driver of IAV with only a small effect from meteorology."

L250-252: Is the simulated IAV in tropospheric CO concentrations driven by biomass burning emissions similar to the IAV in the imposed biomass burning emissions? I would imagine that the IAV in GFED4 CO emissions would be similar to that for CO simulated by the model. Would be useful to confirm this. This then begs the question - what is driving the interannual variability in biomass burning emissions - is it changes in area burnt, biomass available for burning, or meteorological conditions or all of these?

**Authors' response**: Simulated CO IAV is slightly smaller than the IAV of biomass burning emissions from GFEDv4, with a coefficient of variation of 14.3% compared to 20.6%. This

indicates a strong dependence on biomass burning IAV but with additional elements driving tropospheric CO concentrations which limit the control of biomass burning on IAV, such as meteorology and anthropogenic emissions.
The IAV of biomass burning emissions has been investigated in a number of studies with primary drivers including precipitation and temperature (Balzter et al., 2005). Human activity (Achard et al., 2008), ENSO (Hess et al., 2001; Page et al., 2002; van der Werf et al., 2004) and Arctic oscillation (Balzter et al., 2005) have also been proposed as important drivers.

L297-298: I find this sentence confusing - is it that including CO results in a decreasing trend in OH? Though, this is not evident from figure 6.

**Authors' response:** We agree this was confusing. The statement was intended to say that throughout the entire simulation, the effect of CO from fires was to decrease OH concentrations. This is evidenced by negative values for CTRL-COfix in figure 7a (previously figure 6a), indicating that OH was higher when CO emissions from fires are fixed. The text has now been amended to state this more clearly.

**Changes in manuscript:**
**L350-352**
"When CO emissions from biomass burning are fixed, OH concentrations are consistently higher than in the CTRL simulation. This indicates that high CO emissions decrease global mean tropospheric OH."

L316-317: Given that the reaction rate constant of $CH_4+OH$ is strongly sensitive to temperature, approximately 2 % $K^{-1}$ (John et al., 2012), what is the impact of assuming constant temperature in the box model on the results discussed here?

**Authors' response:** The impact of assuming constant temperature or varying temperature in a one-box model is shown in McNorton et al. (2016a) (figure 1c), where the impact was found to be small although not negligible. Therefore, it was decided in this study to utilise a fixed temperature box model as varying temperature has a relatively small impact on the derived methane concentrations. This is now explained in the text.

**Changes in manuscript:**
**L372-373**
"A fixed temperature was used as varying temperature has been found to have a relatively small impact on derived $CH_4$ concentrations (McNorton et al., 2016a)."

L332-L333: The authors mention that they compared the early mean period (1997-2001) with the end period but Figure 7 shows the early period as 1999-2003 without the influence of El Nino. Please revise this sentence.

**Authors' response:** 1999-2003 is correct and this has now been corrected in the text. This was used as a more appropriate mean than 1997-2001, due to the influence of the large El Nino event.

L335: Remove "a".

**Authors' response:** Correction made.

L337-338: The influence of shift in ozone precursor emissions from the mid-latitudes to the tropics has been demonstrated by Zhang et al. (2016).

Thank you for this point. The spatial shift in emissions described in Zhang et al. (2016) is seen in our study by the decreasingly $NO_x$-limited production of $O_3$ in India, but otherwise is not particularly evident over this relatively short period. We have now amended the text citing Zhang et al. (2016) and mention the implications of the shift in the distribution of emissions.

**Changes in manuscript:**
**L395-397**
"This shift in the spatial distribution of $O_3$ precursor emissions to lower latitudes leads to increased tropospheric $O_3$ production proportional to total emissions (Zhang et al., 2016)."

L344: This sentence is confusing. From figure 7b, it looks like India becomes "bothlimited" towards the end of the simulation.

**Authors' response:** We agree. The intention was to state that India does become more "both-limited" at the end of the period, but El Niño conditions seem to inhibit this trend, maintaining the $NO_x$-limited regime over India. The text has now been amended to clarify this:

**Changes in manuscript:**
**L403-404**
"Over India, El Niño conditions inhibit the trend towards a "both-limited" regime, as the $NO_x$-limited regime dominates throughout."

L347-349: Do observations also indicate this significant alteration of the vertical distribution of tropospheric ozone? How does this interpretation of the model results compare with the analysis of Chandra et al (1998) and Ziemke and Chandra (1999)?

**Authors' response:** Thank you for this point. We agree that this discussion is important here. We have now included more detail on the effect of El Nino on tropospheric $O_3$ and comparisons to previous studies. This has been included in section 4.1.

**Changes in manuscript:**
**L410-435**
"The 1997 El Niño significantly altered the vertical distribution of $O_3$ in the troposphere, increasing $O_3$ concentrations in the NH while decreasing in the SH and tropics with an overall decrease in tropospheric $O_3$ of -0.82% compared to the 1997-2014 mean (Fig. 9a). In the CTRL simulation there is decreased $O_3$ in the tropical upper troposphere, possibly related to increased convection over the Eastern Pacific (Oman et al., 2013; Neu et al., 2014). We also simulate large increases in the mid-latitude upper troposphere of both hemispheres in the CTRL and FIREFIX simulations but not in METFIX, implying that this is produced by El Niño-associated meteorological processes which promote intrusion of stratospheric air into the troposphere. These positive anomalies were also observed in Oman et al. (2013) and Zeng and Pyle (2005), attributed to El Niño influence on circulation patterns and enhanced stratospheric-troposphere exchange.

In general, the METFIX run simulates higher $O_3$ concentrations in the NH than the period mean and lower concentrations in the SH (Fig. 9b). This hemispherical shift is also present in the CTRL and FIREFIX simulations but with greater negative $O_3$ anomalies in the SH. The simulated NH increases in the CTRL simulation correspond to other studies of the 1997 El Niño (Koumoutsaris et al., 2008), while Oman et al. (2013) similarly reported negative $O_3$ anomalies in the SH during El Niño. Large increases in tropospheric $O_3$ in the Western Pacific, Indian Ocean and Europe contribute to the increase in $O_3$ in the NH, despite

decreased $O_3$ in the Eastern Pacific (Chandra et al., 1998; Koumoutsaris et al., 2008; Oman et al., 2011).

There is an overall increase in $O_3$ (~2%) when meteorology was fixed to an ENSO-neutral year (i.e. 2013), meaning that meteorology during the 1997 El Niño caused a decrease in tropospheric $O_3$ concentrations despite large increases in $O_3$ in regions of the upper troposphere due to stratospheric intrusion. During the 1997 El Niño we find a 0.4% increase in global tropospheric humidity compared to the period mean. This is likely partly responsible for the general decrease in $O_3$ due to meteorology, as increased humidity enhances $O_3$ loss (Stevenson et al., 2000; Isaksen et al., 2009; Kawase et al., 2011). Changes to transport and distribution of $O_3$ will also impact how efficiently tropospheric $O_3$ is produced and lost.

The similarities between the tropospheric $O_3$ distribution in the CTRL and FIREFIX simulations shows that fire emissions have a relatively small impact on the global distribution of $O_3$, but do affect absolute values, as concentrations in the FIREFIX run are significantly lower at the tropics. This is likely because of the removal of large emissions of $O_3$ precursors in that latitude band when fire emissions are fixed to a non-El Niño year, as several studies have found that enhanced fires in 1997 El Niño increased tropospheric $O_3$ in the region (Chandra et al., 1998; Thompson et al., 2001; Doherty et al., 2006; Oman et al., 2013)."

L385-L387: These are broad-brush statements which then make it easy to question the model's ability to simulate chemical composition during El Nino years and derived interpretations. As I mentioned above, a more focused evaluation would be helpful to reveal model strengths and weaknesses building confidence in the results.

**Authors' response:** To address this point, we have now evaluated the model response to El Niño in section 3, as well as quantifying the model performance (Fig. 1 and S7). The text in the manuscript has now been reworded to reflect this additional analysis.

**Changes in manuscript:**
**L455-459**
"Differences between model and observations may be due to a number of factors, such as the relatively coarse model resolution, uncertainties in the model emission inventories and errors in the observations. However, good overall agreement of the model with different observations, including the ability of the model to simulate the observed atmospheric responses to El Niño events (i.e. OEI change of 2.8 DU compared to 2.4 DU in Ziemke et al. (2010)) provides confidence in model performance and results."

**Responses to reviewer #2:**

General comment: The manuscript "Impact of El Niño Southern Oscillation on the interannual variability of methane and tropospheric ozone" written by Matthew J. Rowlinson describes the impact of meteorological variability and forest fires associated with El Niño Southern Oscillation (ENSO) on methane and tropospheric ozone. The manuscript contains novel investigation to quantitatively isolate the impact of forest fire emissions and meteorological variability on methane lifetime and growth rate using modelling approach. While many modelling studies on the impact of ENSO on tropospheric ozone have been conducted, this is an interesting work that deduced spatial variations in radiative effects of tropospheric ozone changes during El Niño. The authors used appropriate model simulations to use for this science problem. Overall this manuscript is well written and easy to follow. I would like to consider the publication of this manuscript from Atmospheric Chemistry and Physics after minor revision. Please see the following comments.

**Authors' response:** We would like to thank the review for their general comments on the manuscript, suggestions for improvement and positive remarks on the study. We have endeavoured to address all specific comments and our responses and corrections are detailed below.

Specific comments:

1. Model evaluation

In Section 3, the authors presented the model evaluation of mean concentration fields of O3, CO, CH4, and NOx with satellite, aircraft, and ozonesonde observations, whereas the authors do not conduct model evaluation of inter-annual variations in these concentration fields during El Niño. The model evaluation would also be helpful to support validity of the model simulations used in this study.

**Authors' response:** We agree that this is an important point which was also raised by Reviewer #1. We now clarify in section 2.1 that our model is driven by ECMWF-reanalysis, which has previously been shown to be able to represent ENSO events.

**Changes in manuscript:**
**L133-135**
"ECMWF ERA-Interim reanalyses have been shown to have good skill in capturing Madden-Julian Oscillation (MJO) events which in turn impact the onset of ENSO events (Dee et al., 2011), giving confidence that the model competently represents El Niño meteorological conditions."

We have also compared simulated tropospheric $O_3$ response to El Niño events against observed responses in literature. We find that the Ozone ENSO Index (OED) from Ziemke et al. (2010) yields a response of +2.8 DU in the model compared to a +2.4 DU response in observations. We have also added a figure (Fig. S4) to the supplementary showing regional response in total $O_3$ column and compared this to findings of (Zhang et al., 2015). We have consequently changed the manuscript text in section 3:

**Changes in manuscript:**
**L197-204**
"We have also assessed the capability of TOMCAT-GLOMAP to simulate observed responses to El Niño events. Ziemke et al. (2010) derived an $O_3$ ENSO index using satellite observations, finding that for a +1K change in the Nino 3.4 index, there was a 2.4 DU increase in the OEI. In TOMCAT-GLOMAP, we calculate a 2.8 DU increase per +1K in the

Nino 3.4, indicating a slightly larger but comparable response to El Niño events. The regional response of tropospheric $O_3$ to El Niño was evaluated against an analysis using various observations and a chemistry-climate model in Zhang et al. (2015). The study observed increased total $O_3$ column in the North Pacific, southern USA, north-eastern Africa and East Asia, with decreases over central Europe and the North Atlantic. All of these observed responses were present in TOMCAT-GLOMAP simulations, except with a slight increase in central Europe and a simulated decrease in TOC in Western Europe and East Atlantic (Fig. S5). "

**2. Impact of ENSO on OH and methane**

In Section 4.2, the authors quantified the impact of forest fire emissions on CH4 growth rate; however, the authors do not compare it with the observed CH4 growth rate, even though the authors concluded that "This effect, combined with concurrent direct CH4 emission from fires explain the observed changes to CH4 growth rate during the 1997 El Niño" (P. 15, L. 391-392) in Section 6. I would like to recommend to add the comparison of the observed and simulated CH4 growth rate in Section 4.2. The authors also presented the impacts of forest fire emissions and meteorological variability on OH concentrations in Figure 6. Although the impacts are comparable except during the 1997-1998 El Niño, the authors hardly mentioned the impact of meteorology. A more detailed analysis (e.g., how do specific humidity, cloud, and lightning NOx affect inter-annual variations in OH?) would also be interesting, though this may be beyond the scope of the current manuscript.

**Authors' response:** We thank the reviewer for this very good point and useful recommendations. Our main reason for using the box model was to analyse the effect of modelled changes to OH under different scenarios. Therefore, in our study, the difference in the changes to methane concentrations and growth rates between different simulations are more relevant than comparing the absolute methane concentrations to observed values. It is important to note that the box model only takes into account changes to $CH_4$ due to loss by OH, whereas observed changes are impacted by all sink and source variabilities, making the comparison difficult to make.
However, we have now also compared the simulated methane growth rate from the box model with the observed growth rate (figure 1 below). We find that in general the box model reproduces the broad trend in growth rates in the early part of the period, and in particular picks up the large change in 1998, caused by the El Niño event analysed in this study. Later in the modelled period the comparison with the observed growth rate becomes poorer.

[Figure]

*Figure 1. Comparison of global mean CH₄ concentration growth rate from the box model to the (Dlugokencky, 2019) observed values.*

Regarding OH variability due to meteorology, we have now added to our discussion to consider the causes of this. Various studies have found that meteorology is important for OH variability. Nicely et al. (2018) quantified the impact of stratospheric $O_3$, tropospheric $H_2O_2$ and $NO_x$, temperature and circulation changes on global OH, with $H_2O_2$ and $NO_x$ changes being the most important. Lightning $NO_x$ has been found to be important for OH in Turner et al. (2018), while Murray et al. (2014) examined variability on longer timescales, finding multiple meteorological drivers.

We have now modified the manuscript as follows:

**Changes in manuscript:**
**L324-338**
"Various meteorological variables are known to affect OH and $O_3$ variability, including humidity, clouds and temperature (Stevenson et al., 2005; Holmes et al., 2013; Nicely et al., 2018). OH variability is particularly sensitive to changes in lightning $NO_x$ production which decreases during El Niño conditions (Turner et al., 2018). Murray et al. (2014) also examined factors affecting OH variability since the last glacial maximum, finding tropospheric water vapour, overhead stratospheric $O_3$ and lightning $NO_x$ to be key controlling factors. Furthermore, circulation changes during El Niño events have been linked to lower stratospheric $O_3$ variability (Zhang et al., 2015; Manatsa and Mukwada, 2017), which in turn influences tropospheric OH and $O_3$ concentrations (Holmes et al., 2013; Murray et al., 2014). Despite the importance of meteorological drivers, we find that fire emissions are the dominant cause of variation in both OH and $O_3$ during the 1997 El Niño, increasing global tropospheric $O_3$ burden by up to ~7% and decreasing tropospheric OH by up to ~6%. This result is supported by several other studies, which have found that during large fire events such as that caused by the 1997 El Niño, fire emissions substantially decrease tropospheric OH and increase tropospheric $O_3$ (Hauglustaine et al., 1999; Sudo and Takahashi, 2001; Holmes et al., 2013). Our results indicate that while meteorology is generally the most important driver of IAV in global tropospheric OH and $O_3$, fire emissions can also play a key role and become the dominant driver when there are particularly large fire emissions related to El Niño. "

P. 2, L. 57: Tropospheric O3 changes related to meteorology were attributed by both transport and O3 loss with water vapour in Section 4.3; why do the authors mention only atmospheric transport changes in abstract?

Authors' response: Thank you for pointing this out. We have now modified this sentence in the abstract to better account for the likely pathways of decreased tropospheric $O_3$ during El Niño.

**Changes in manuscript:**
**L58-59**

". El Niño-related changes in atmospheric transport and humidity decrease global tropospheric $O_3$ concentrations leading to a -0.03 $Wm^{-2}$ change in the $O_3$ radiative effect (RE). "

P. 4, L. 139-140: Why is CH4 concentrations scaled to the observations even though the CH4 emission inventory is used in the model?

**Authors' response:** We agree this requires clarification. The scaling in TOMCAT applies only to the global mean surface methane, meaning that the distribution of methane, which is simulated in the model, still needs to be based on an emissions inventory. An improved explanation of the $CH_4$ scaling applied in TOMCAT is now included in section 2.

**Changes in manuscript:**
**L146-148**
"Global mean surface $CH_4$ mixing ratio is scaled in TOMCAT-GLOMAP to a best-estimate based on observed global surface mean concentration (McNorton et al., 2016a; Dlugokencky, 2019)."

P. 8, L. 242-243: What is definition of El Niño periods used in this study? Please clarify the definition.

**Authors' response:** El Nino periods were defined to be times when the bimonthly multivariate ENSO index was greater than 1. We have now included this important clarification in section 2.3.

**Changes in manuscript:**
**L166**
"Throughout this study, an El Niño event was taken to be ongoing if the bimonthly MEI was greater than +1.0."

P. 8, L. 253-255: I would like to recommend to add the percentage number of the increase during the 2002/2003, 2006, and 2009/2010 El Niño events to compare them with the extreme El Niño event in 1997/1998 quantitatively.

**Authors' response**: Thank you for this point - we agree that this more quantitative statement provides a clearer picture. We have now added these values and amended the text.

**Changes in manuscript:**
**L289-291**

"Smaller increases of 5.8% and 7.6% occur during less extreme El Niño events of 2002/2003 and 2006, respectively, with only a 1.8% increase during the 2009/2010 El Niño, indicating that El Niño only significantly impacts CO concentrations when there is an associated increase in global fire events."

P. 9, L. 271-272: Why does fire emissions have small impact on CO IAV?

**Authors' response:** In specific regions and seasons meteorology can be more important due to increased convective transport, but in general we report that fire emissions have a large effect on IAV.
This specific sentence stated that meteorology was more important in Africa (Sep-Oct) and Indonesia (Mar-Apr). We go on to say that these results have also been found in other studies. We have now added the following to the text to clarify possible reasons for the stronger meteorological effect on specific regions and seasons.

**Changes in manuscript:**
**L312-315**
 "This is in good agreement with Voulgarakis et al. (2015) who found that with fixed biomass burning emissions there remained high IAV over Africa during Dec-Jan, and Huang et al. (2014) who found CO over Central Africa correlated more closely with ice water content than CO emissions due to increased convective transport"

P. 11, L. 300-302: Do you have possible explanation of the difference between this study and Butler et al. (2005)? Did you compare increases in CO emissions in GFED4 with Butler et al. (2005)?

**Authors' response:** We agree this discrepancy needed further clarification. The 9% figure given in our manuscript was a maximum monthly OH perturbation. When we average our results over the same period as Butler et al. (2005), we find a mean perturbation of -3.6%. While still slightly larger, this is much closer to the -2.2% value reported in Butler et al. (2005) and also compares well to modelled OH change due to the 1997 Indonesian wildfires in Duncan et al. (2003), of between -2.1% and -6.8%. We have amended the text in section 4.2.

**Changes in manuscript:**
**L352-356**
"The greatest impact is during the 1997 El Niño where CO emissions were abnormally large, suppressing mass weighted global monthly mean OH concentrations by up to 9%. The mean effect on OH over the 1997 El Niño of -3.6% is comparable to that simulated by Butler et al. (2005), who also found an increase in CO resulted in achange in OH of -2.2%. Duncan et al. (2003) found a similar magnitude response in OH to the Indonesian wildfires in 1997, of between -2.1% and -6.8%."

P. 13, L. 327: Please clarify the reason why you conducted this analysis in Section 4.3.

**Authors' response:** We have now included justification for this analysis at the start of section 4.3.

**Changes in manuscript:**
**L382-384**
"In this section we examine trends and the impact of El Niño on the production of tropospheric $O_3$. El Niño is known to have large effects on tropospheric $O_3$ precursors such as CO and $NO_x$, therefore examining $O_3$ production regimes during El Nino can provide insights into the main mechanism responsible for the observed changes in tropospheric $O_3$."

P. 14, L. 346-362: Many previous works have been done with different models and satellite observations with regard to ENSO impacts on tropospheric ozone. It would be beneficial to see more discussion of how the results presented here compare to previous studies (e.g., Stevenson et al., 2005; Zeng and Pyle, 2005; Doherty et al., 2006; Koumoutsaris et al., 2008; Nassar et al., 2009; Ziemke et al., 2010; Sekiya and Sudo, 2012; Oman et al., 2013; Neu et al., 2014; Inness et al., 2015).

**Authors' response:** Thank you for this point. We agree that the limited discussion of El Nino and tropospheric $O_3$ was a weakness of the original manuscript. We have now added substantially enhanced this discussion, including comparison to observations and previous studies, clarifying how our findings relate and offering improved explanations for the changes in our simulations.

**Changes in manuscript:**
**L410-435**
The 1997 El Niño significantly altered the vertical distribution of $O_3$ in the troposphere, increasing $O_3$ concentrations in the NH while decreasing in the SH and tropics with an overall decrease in tropospheric $O_3$ of -0.82% compared to the 1997-2014 mean (Fig. 9a). In the CTRL simulation there is decreased $O_3$ in the tropical upper troposphere, possibly related to increased convection over the Eastern Pacific (Oman et al., 2013; Neu et al., 2014). We also simulate large increases in the mid-latitude upper troposphere of both hemispheres in the CTRL and FIREFIX simulations but not in METFIX, implying that this is produced by El Niño-associated meteorological processes which promote intrusion of stratospheric air into the troposphere. These positive anomalies were also observed in Oman et al. (2013) and Zeng and Pyle (2005), attributed to El Niño influence on circulation patterns and enhanced stratospheric-troposphere exchange.

In general, the METFIX run simulates higher $O_3$ concentrations in the NH than the period mean and lower concentrations in the SH (Fig. 9b). This hemispherical shift is also present in the CTRL and FIREFIX simulations but with greater negative $O_3$ anomalies in the SH. The simulated NH increases in the CTRL simulation correspond to other studies of the 1997 El Niño (Koumoutsaris et al., 2008), while Oman et al. (2013) similarly reported negative $O_3$ anomalies in the SH during El Niño. Large increases in tropospheric $O_3$ in the Western Pacific, Indian Ocean and Europe contribute to the increase in $O_3$ in the NH, despite decreased $O_3$ in the Eastern Pacific (Chandra et al., 1998; Koumoutsaris et al., 2008; Oman et al., 2011).

There is an overall increase in $O_3$ (~2%) when meteorology was fixed to an ENSO-neutral year (i.e. 2013), meaning that meteorology during the 1997 El Niño caused a decrease in tropospheric $O_3$ concentrations despite large increases in $O_3$ in regions of the upper troposphere due to stratospheric intrusion. During the 1997 El Niño we find a 0.4% increase in global tropospheric humidity compared to the period mean. This is likely partly responsible for the general decrease in $O_3$ due to meteorology, as increased humidity enhances $O_3$ loss (Stevenson et al., 2000; Isaksen et al., 2009; Kawase et al., 2011). Changes to transport and distribution of $O_3$ will also impact how efficiently tropospheric $O_3$ is produced and lost.

The similarities between the tropospheric $O_3$ distribution in the CTRL and FIREFIX simulations show that fire emissions have a relatively small impact on the global distribution of $O_3$, but do affect absolute values, as concentrations in the FIREFIX run are significantly lower at the tropics. This is likely because of the removal of large emissions of $O_3$ precursors in that latitude band when fire emissions are fixed to a non-El Niño year, as several studies

have found that enhanced fires in 1997 El Niño increased tropospheric $O_3$ in the region (Chandra et al., 1998; Thompson et al., 2001; Doherty et al., 2006; Oman et al., 2013)."

P. 14, L. 364-365: Do the authors used TES O3 radiative kernel? Please clarify what data the authors used in this study.

**Authors' response:** We agree that more clarity is needed here. We used the tropospheric $O_3$ radiative kernel produced by Rap et al. (2015) using the Edwards-Slingo offline radiative transfer model. We have amended the text in section 2.2 to make this clear.

**Changes in manuscript:**
**L150-151**
"Radiative effects of $O_3$ changes are calculated using the $O_3$ radiative kernel approach derived by Rap et al. (2015) using an offline version of the Edwards and Slingo (1996) radiative transfer model."

**Technical corrections:**

P. 2. L. 58: typo for nitrogen oxides?

**Authors' response**: Thank you for pointing this. Correction made.

P. 4, L. 135: typo for nitrogen oxides?

**Authors' response:** Correction made.

P. 13, L. 332: Which is correct, "1997-2001" or "1999-2003"? The "1999-2003" period would be appropriate to obtain the mean fields.

**Authors' response**: Thank you for pointing this. 1999-2003 is correct and this has now been corrected in the text. This was used as a more appropriate mean than 1997-2001, due to the influence of the large El Nino event.

**Changes in manuscript:**
**L390-391**
"We compare the early period mean (1999-2003) to the end period mean (2010-2014) to determine whether significant changes have occurred over the 18-year period, and compared mean El Niño conditions to both."

[revised manuscript text omitted]